# What Matters in Data for DPO?

Yu Pan[1] *    Zhongze Cai[2]    Guanting Chen[3]    Huaiyang Zhong[4]    Chonghuan Wang[5]

[1] University of Sydney    [2]Imperial College London    [3]University of North Carolina at Chapel Hill
[4] Virginia Tech    [5] University of Texas at Dallas

## Abstract

Direct Preference Optimization (DPO) has emerged as a simple and effective approach for aligning large language models (LLMs) with human preferences, bypassing the need for a learned reward model. Despite its growing adoption, a fundamental question remains open: what characteristics of preference data are most critical for DPO performance? In this work, we provide a systematic study of how preference data distribution influences DPO, from both theoretical and empirical perspectives. We show that the quality of chosen responses plays a dominant role in optimizing the DPO objective, while the quality of rejected responses may have relatively limited impact. Our theoretical analysis characterizes the optimal response distribution under DPO and reveals how contrastiveness between responses helps primarily by improving the chosen samples. We further study an online DPO setting and show it effectively reduces to supervised fine-tuning on the chosen responses. Extensive experiments across diverse tasks confirm our findings: improving the quality of chosen responses consistently boosts performance regardless of the quality of the rejected responses. We also investigate the benefit of mixing the on-policy data. Our results interpret the mechanism behind some widely adopted strategies and offer practical insights for constructing high-impact preference datasets for LLM alignment.

## 1  Introduction

The importance of aligning large language models (LLMs) with human preferences cannot be overstated. Two leading paradigms for achieving this alignment are Reinforcement Learning from Human Feedback (RLHF) (Bai et al., 2022; Ouyang et al., 2022) and Direct Preference Optimization (DPO) (Rafailov et al., 2023). The key difference lies in whether an explicit reward model is trained (as in RLHF) or whether the model itself is optimized directly using preference data (as in DPO). Significant effort has been devoted to improving the performance of these methods by constructing more effective preference datasets. Common techniques include rejection sampling (i.e., generating multiple responses and selecting the best or worst, see Khaki et al. 2024), annotator rewriting/editing, and iterative use of on-policy data (Tajwar et al., 2024).

However, despite the empirical progress, fundamental questions about *what properties of preference data actually matter for alignment* remain underexplored. For example: Do chosen and rejected responses contribute symmetrically during optimization? How does the contrastiveness between response pairs affect learning? Under what conditions does incorporating on-policy data lead to gains?

In this paper, we provide a systematic study of the role of preference data in DPO, combining theoretical analysis with empirical validation. We begin by analyzing how the preference dataset's coverage of the high quality responses influences the gradient of the DPO loss. We begin by examining how the coverage of high-quality responses in the preference dataset influences the gradient

---

*Corresponding to: yu.pan@sydney.edu.au

39th Conference on Neural Information Processing Systems (NeurIPS 2025).

of the DPO loss. Insufficient coverage of such responses can hinder optimization, as the DPO objective lacks an explicit gradient signal to promote high-reward outputs when they are absent from the comparisons in $\mathcal{D}_{\text{DPO}}$. Then, we analyze the optimal distribution that minimizes the DPO loss and how it is shaped by the distribution of the preference dataset. Our theoretical results show that the quality of the chosen responses plays a dominant role in DPO performance, while the quality of rejected responses has a more limited impact. We further demonstrate that the widely adopted strategy of increasing contrastiveness between responses is effective primarily because it tends to elevate the quality of the chosen responses. Moreover, we examine a simplified online DPO setting in which high-quality chosen responses remain fixed, and rejected responses are generated in an online fashion. We show that this setting essentially reduces to supervised fine-tuning on the chosen responses, further highlighting the central role of the quality of the chosen responses.

We empirically validate our theoretical insights across multiple tasks and datasets. When fixing the chosen responses and varying the quality of the rejected ones, we observe little change in DPO performance. In contrast, when fixing the rejected responses and increasing the quality of the chosen ones, DPO performance consistently improves. Additionally, when holding the quality gap between chosen and rejected responses constant, improving the absolute quality of the chosen responses leads to better outcomes. Finally, we investigate how mixing on-policy and offline data affects performance under varying levels of offline data quality.

## 2 Preliminaries

**Supervised Fine-tuning (SFT).** SFT is is typically the first stage in adapting a pre-trained LLM to downstream tasks. Given a dataset $\mathcal{D}_{\text{SFT}}$ consisting of high-quality instruction-response pairs $(\mathbf{x}, \mathbf{y})$ (Ouyang et al. 2022), the objective is to maximize the log-likelihood of the the demonstration data. Specifically, SFT is minimizing the loss function:

$$\mathcal{L}_{\text{SFT}}(\boldsymbol{\theta}; \mathcal{D}_{\text{SFT}}) = -\mathbb{E}_{(\mathbf{x},\mathbf{y}) \sim \mathcal{D}_{\text{SFT}}}[\log \pi_{\boldsymbol{\theta}}(\mathbf{y}|\mathbf{x})]. \tag{1}$$

**Reinforcement Learning from Human Feedback (RLHF).** After SFT, fine-tuning with human preference data is widely used to further align the model. RLHF begins by training a reward model $r(\mathbf{x}, \mathbf{y})$ to reflect human preferences based on a preference dataset $\{(\mathbf{x}_i, \mathbf{y}_{i.w}, \mathbf{y}_{i,l})_{i \geq 1}\}$, where $\mathbf{y}_{i,w}$ and $\mathbf{y}_{i,l}$ denote the preferred and rejected response, respectively. The policy $\pi_{\theta}$ is then optimized, typically using reinforcement learning algorithms such as PPO (Schulman et al. 2017, Ouyang et al. 2022), by minimizing the following objective:

$$\mathcal{L}_{\text{RLHF}}(\boldsymbol{\theta}) = -\mathbb{E}_{\mathbf{x} \sim \mathcal{D}_{\mathbf{x}}, \mathbf{y} \sim \pi_{\boldsymbol{\theta}}(\cdot|\mathbf{x})}[r(\mathbf{x}, \mathbf{y})] + \beta \mathbb{D}_{\text{KL}}(\pi_{\boldsymbol{\theta}}(\mathbf{y}|\mathbf{x})||\pi_{\text{ref}}(\mathbf{y}|\mathbf{x})). \tag{2}$$

**Directed Preference Optimization (DPO).** To simplify the process of RLHF, particularly to get rid of the training of a reward model, Rafailov et al. (2023) realize that the reward function can be represented by the learning policy:

$$r_{\boldsymbol{\theta}}(\mathbf{x}, \mathbf{y}) = \beta[\log \pi_{\boldsymbol{\theta}}(\mathbf{y}|\mathbf{x}) - \log \pi_{\text{ref}}(\mathbf{y}|\mathbf{x})] + \beta \log Z_{\boldsymbol{\theta}}(\mathbf{x}), \tag{3}$$

where $Z(\mathbf{x}) = \sum_{\mathbf{y}} \pi_{\text{ref}}(\mathbf{y}|\mathbf{x}) \exp(r_{\boldsymbol{\theta}}(\mathbf{x}, \mathbf{y})/\beta)$ is the partition function. Based on the Bradley-Terry (BT) preference assumption (Bradley and Terry 1952), together with the pairs of chosen and rejected responses, DPO fine-tunes the language model by optimizing the following loss function:

$$\mathcal{L}_{\text{DPO}}(\boldsymbol{\theta}; \mathcal{D}_{\text{DPO}}) = -\mathbb{E}_{(\mathbf{x},\mathbf{y}_w,\mathbf{y}_l) \sim \mathcal{D}_{\text{DPO}}} \left[\log \sigma \left(\beta \log \frac{\pi_{\boldsymbol{\theta}}(\mathbf{y}_w|\mathbf{x})}{\pi_{\text{ref}}(\mathbf{y}_w|\mathbf{x})} - \beta \log \frac{\pi_{\boldsymbol{\theta}}(\mathbf{y}_l|\mathbf{x})}{\pi_{\text{ref}}(\mathbf{y}_l|\mathbf{x})}\right)\right], \tag{4}$$

where $\mathbf{y}_w$ and $\mathbf{y}_l$ are the chosen response and the rejected response respectively, and $\sigma(x) = \frac{1}{1+e^{-x}}$. The global minimizer of Eqs. (2) and (4) under BT assumption has been well understood to be

$$\pi_{r^*}(\mathbf{y}|\mathbf{x}) \propto \pi_{\text{ref}}(\mathbf{y}|\mathbf{x}) \exp\left(\frac{1}{\beta} r^*(\mathbf{x}, \mathbf{y})\right), \tag{5}$$

where $r^*$ is the true reward model. Rafailov et al. (2023) also derive the derivative of the DPO loss (4) with respect to the parameters $\boldsymbol{\theta}$:

$$\nabla_{\boldsymbol{\theta}} \mathcal{L}_{\text{DPO}}(\boldsymbol{\theta}; \mathcal{D}_{\text{DPO}}) =$$
$$- \beta \mathbb{E}_{(\mathbf{x},\mathbf{y}_w,\mathbf{y}_l) \sim \mathcal{D}_{\text{DPO}}} \left[\sigma(\hat{r}_{\boldsymbol{\theta}}(\mathbf{x}, \mathbf{y}_l) - \hat{r}_{\boldsymbol{\theta}}(\mathbf{x}, \mathbf{y}_w)) \left[\nabla_{\boldsymbol{\theta}} \log \pi_{\boldsymbol{\theta}}(\mathbf{y}_w|\mathbf{x}) - \nabla_{\boldsymbol{\theta}} \log \pi_{\boldsymbol{\theta}}(\mathbf{y}_l|\mathbf{x})\right]\right], \tag{6}$$

where $\hat{r}_{\boldsymbol{\theta}}(\mathbf{x}, \mathbf{y}) = \beta \log \frac{\pi_{\boldsymbol{\theta}}(\mathbf{y}|\mathbf{x})}{\pi_{\text{ref}}(\mathbf{y}|\mathbf{x})}$ denotes the implicit reward induced by the language model $\pi_{\boldsymbol{\theta}}$ and the reference model $\pi_{\text{ref}}$.

In this work, we aim to better understand the role of the data distribution $\mathcal{D}_{\text{DPO}}$, and identify the key factors that contribute to successful DPO training.

## 3  Related Works

**RL-based LLM Alignment and DPO.** Following SFT, RL policy gradient methods are then employed to align model outputs with human preferences encoded through reward modeling. While early RL approaches like TRPO (Schulman et al., 2015) and PPO (Schulman et al., 2017) established foundational frameworks, their computational intensity motivated the development of resource-efficient alternatives such as RAFT (Dong et al., 2023), RRHF (Yuan et al., 2023), SLiC (Zhao et al., 2023), and ORPO (Hong et al., 2024). DPO (Rafailov et al., 2023) cleverly reinterprets the RL objective through contrastive loss by eliminating explicit reward value while maintaining stable training dynamics with reduced computational demands. The DPO framework has subsequently inspired multiple variants including KTO (Ethayarajh et al., 2024), IPO (Azar et al., 2024), and CPO (Xu et al., 2024a). Recent work by Shao et al. (2024) attempts to unify alignment-stage training paradigms through a generalized perspective.

**Data Quality in LLM Alignment.** The critical role of data quality in LLM alignment has been rigorously established across training paradigms. Early studies (Zhou et al., 2023a) demonstrated its decisive impact in fine-tuning contexts, while the contemporary focus on reasoning (Muennighoff et al., 2025) further underscores the performance gains attainable through carefully curated alignment data. Empirical evidence specifically in DPO training (Morimura et al., 2024; Wu et al., 2024; Ivison et al., 2024) reveals two critical insights: (1) DPO exhibits stronger sensitivity to data quality compared to traditional RL methods like PPO; (2) strategic selection of high-quality samples improve DPO training performance. While Khaki et al. (2024) and Gou and Nguyen (2024) suggest larger preference gaps improve DPO, Pattnaik et al. (2024) and Xiao et al. (2025) find moderate gaps beneficial. However, a systematic understanding of the role of data quality remains lacking in the literature.

**On-policy DPO.** Another fruitful stream of literature that is related to the proper usage of data is on-policy DPO implementations (Yuan et al., 2024; Chen et al., 2024; Guo et al., 2024; Rosset et al., 2024; Tajwar et al., 2024; Pang et al., 2024). Empirical analyses by Xu et al. (2024b) reveal that distributional mismatch between training data and the base model's original domain disproportionately impacts DPO compared to PPO. On-policy DPO actively samples the intermediate model generations, which serves as an adaptive distributional bridge to mitigate out-of-domain degradation. Despite the potential benefit of on-policy DPO, excessive reliance on the on-policy data is very likely to induce training instability that can lead to a significant drop in model performance (Lambert et al., 2024; Deng et al., 2025). Feng et al. (2025) propose PILAF, a theoretically-grounded sampling strategy for online and iterative DPO, which shares our work's conceptual focus on attributing optimization signals to the data distribution. Their method uses policy interpolation to explicitly align the training gradient with the true oracle objective and help stablize the training process. While trials have also been made to balance the on-policy and off-policy data integration (Wang et al., 2025), understanding when and how on-policy data can be helpful also remains to be further explored.

## 4  DPO Interpretation

In this section, we provide theoretical insights into what characteristics of a dataset matter most for DPO performance, and explain why some widely used data generation strategies are effective.

### 4.1  The Role of Distributions of Chosen and Rejected Samples

We begin by analyzing the standard DPO setup, where $\mathcal{D}_{\text{DPO}}$ is generated in two steps. First, a triplet $(\mathbf{x}, \mathbf{y}_1, \mathbf{y}_2)$ is sampled from the distribution $\mathcal{D}_u = \mathcal{X} \times \mathcal{Y}_1 \times \mathcal{Y}_2$. Then, a preference label is assigned by the BT model, which identifies the more preferred response as $\mathbf{y}_w$ and the less preferred one as $\mathbf{y}_l$. One important perspective we want to highlight for understanding the role of $\mathcal{D}_{\text{DPO}}$ is coverage. Specifically, let us consider the classical solution Eq. (5) which is heavily decided by the optimal

reward model $r^*$. When our $\mathcal{D}_{\text{DPO}}$ fail to include examples representative of responses with high true rewards in terms of $r^*$, then this optimal reward $r^*$ may not be identifiable, especially in the regime of "high reward", from the data alone. This lack of coverage can complicate the optimization process, as there is no explicit gradient signal within the DPO objective to increase the likelihood of high-reward responses if they are absent from the preference comparisons in $\mathcal{D}_{\text{DPO}}$.

To formalize this, let us fix a prompt $\mathbf{x}$ and focus on a high-reward response $\mathbf{y}_h$, i.e., $r^*(\mathbf{x}, \mathbf{y}_h)$ is high. For a generating policy $\pi_{\boldsymbol{\theta}}(\cdot|\mathbf{x})$, a higher value of $\pi_{\boldsymbol{\theta}}(\mathbf{y}_h|\mathbf{x})$ corresponds to a greater likelihood of generating high-quality responses. Intuitively, if $\mathbf{y}_h$ is not covered by the dataset, DPO has no mechanism to increase its likelihood. The following theorem

**Theorem 4.1.** *Let $\pi_{\boldsymbol{\theta}_t}$ be the policy trained with gradient descent on the DPO loss* (4) *at step $t$ under the preference data $\mathcal{D}_{DPO}$. Then for a given high-reward response $(\mathbf{x}, \mathbf{y}_h)$, the likelihood $\pi_{\boldsymbol{\theta}_{t+1}}(\mathbf{y}_h|\mathbf{x})$ will not change if $\mathbf{y}_h$ is not in the support of $\mathcal{D}_u$. That is*

$$\pi_{\boldsymbol{\theta}_{t+1}}(\mathbf{y}_h|\mathbf{x}) = \pi_{\theta_t}(\mathbf{y}_h|\mathbf{x}) \quad \text{if } \mathbf{y}_h \notin supp(\mathcal{D}_u).$$

Theorem 4.1 reveals that if $\mathbf{y}_h$ is not in the support of $\mathcal{D}_u$, $\pi_{\boldsymbol{\theta}_t}$ will not get the signal to converge towards the high quality response. Therefore, without sufficient coverage, DPO cannot promote desirable behaviors, regardless of how well the loss is minimized. In practice, this suggests that data selection and filtering strategies should not only focus on clear preferences but also ensure that high-reward responses are adequately represented to enable generalization. Following Theorem 4.1, if we have a closer investigation on when the $\pi_{\boldsymbol{\theta}_t}(\mathbf{y}|\mathbf{x})$ changes, the following proposition provides a formal characterization of how the DPO training process updates the likelihood of responses, depending on how the current models preference ranking aligns with the true preferences, which may be of independent interest.

**Proposition 4.2.** *Following the notation of Theorem 4.1, if $\mathbf{y}_h \in supp(\mathcal{D}_u)$, the change in likelihood $\pi_{\theta_{t+1}}(\mathbf{y}_h|\mathbf{x})$ satisfies:*

   *(i)* $\pi_{\boldsymbol{\theta}_{t+1}}(\mathbf{y}_h|\mathbf{x}) > \pi_{\theta_t}(\mathbf{y}_h|\mathbf{x})$ *if* $\bar{\mathbb{P}}_{Y_2 \sim \mathcal{D}_u|\mathbf{x},\mathbf{y}_h}(\mathbf{y}_h \succ Y_2|\mathbf{x}) > \bar{\mathbb{P}}^{BT(\boldsymbol{\theta}_t)}_{Y_2 \sim \mathcal{D}_u|\mathbf{x},\mathbf{y}_h}(\mathbf{y}_h \succ Y_2|\mathbf{x})$;

   *(ii)* $\pi_{\boldsymbol{\theta}_{t+1}}(\mathbf{y}_h|\mathbf{x}) = \pi_{\theta_t}(\mathbf{y}_h|\mathbf{x})$ *if* $\bar{\mathbb{P}}_{Y_2 \sim \mathcal{D}_u|\mathbf{x},\mathbf{y}_h}(\mathbf{y}_h \succ Y_2|\mathbf{x}) = \bar{\mathbb{P}}^{BT(\boldsymbol{\theta}_t)}_{Y_2 \sim \mathcal{D}_u|\mathbf{x},\mathbf{y}_h}(\mathbf{y}_h \succ Y_2|\mathbf{x})$;

   *(ii)* $\pi_{\boldsymbol{\theta}_{t+1}}(\mathbf{y}_h|\mathbf{x}) < \pi_{\theta_t}(\mathbf{y}_h|\mathbf{x})$ *if* $\bar{\mathbb{P}}_{Y_2 \sim \mathcal{D}_u|\mathbf{x},\mathbf{y}_h}(\mathbf{y}_h \succ Y_2|\mathbf{x}) < \bar{\mathbb{P}}^{BT(\boldsymbol{\theta}_t)}_{Y_2 \sim \mathcal{D}_u|\mathbf{x},\mathbf{y}_h}(\mathbf{y}_h \succ Y_2|\mathbf{x})$,

*where $\bar{\mathbb{P}}_{Y_2 \sim \mathcal{D}_u|\mathbf{x},\mathbf{y}_h}(\mathbf{y}_h \succ Y_2|\mathbf{x})$ denotes $\mathbb{E}_{Y_2 \sim \mathcal{D}_u|(X,Y_1)=(\mathbf{x},\mathbf{y}_h)}[\mathbb{P}(\mathbf{y}_h \succ Y_2|\mathbf{x})]$, $\bar{\mathbb{P}}^{BT(\boldsymbol{\theta}_t)}_{Y_2 \sim \mathcal{D}_u|\mathbf{x},\mathbf{y}_h}(\mathbf{y}_h \succ Y_2|\mathbf{x})$ represents $\mathbb{E}_{Y_2 \sim \mathcal{D}_u|(X,Y_1)=(\mathbf{x},\mathbf{y}_h)}[\mathbb{P}^{BT}_{\boldsymbol{\theta}_t}(\mathbf{y}_h \succ Y_2|\mathbf{x})]$, $\mathbb{P}(\mathbf{y}_h \succ Y_2|\mathbf{x})$ is the true probability the $\mathbf{y}_h$ is more preferable, $\mathbb{P}^{BT}_{\boldsymbol{\theta}}$ stands for the BT model parametrized by the current model $\boldsymbol{\theta}$ and $\mathcal{D}_u|X, Y_1$ denotes the conditional distribution of $Y_2$ given $X$ and $Y_1$ under $\mathcal{D}_u$.*

In Proposition 4.2, $\bar{\mathbb{P}}_{Y_2 \sim \mathcal{D}_u|\mathbf{x},\mathbf{y}_h}(\mathbf{y}_h \succ Y_2|\mathbf{x})$ measures the average preference probability for $\mathbf{y}_h$ observed in the dataset under the true preference model, and $\bar{\mathbb{P}}^{\text{BT}(\boldsymbol{\theta}_t)}_{Y_2 \sim \mathcal{D}_u|\mathbf{x},\mathbf{y}_h}(\mathbf{y}_h \succ Y_2|\mathbf{x})$ stands for the preference probability predicted by the current model. Proposition 4.2 reveals when $\mathbf{y}$ is present in the dataset and the current model underestimates how good $\mathbf{y}_h$ is relative to alternatives, DPO will increase its likelihood. Conversely, if the model overestimates $\mathbf{y}_h$'s quality, it will receive a negative update, decreasing its likelihood.

Up till now, the above analysis, as well as many classical results in the literature, relies on the assumption that the dataset $\mathcal{D}_{\text{DPO}}$ satisfies the BT model. However, in practice, we are often provided with a preference dataset without the ability to verify whether this assumption holds. In the following, we provide another perspective on directly analyzing the distribution that minimizes the DPO loss, regardless of whether the BT assumption is valid. We denote the marginal distribution of the chosen response $\mathbf{y}_w$ and the rejected response $\mathbf{y}_l$ of $\mathcal{D}_{\text{DPO}}$ as $\pi_w(\cdot|\mathbf{x})$ and $\pi_l(\cdot|\mathbf{x})$ respectively. For simplicity, we assume that $\mathbf{y}_w$ and $\mathbf{y}_l$ are independently drawn from $\pi_w(\cdot|\mathbf{x})$ and $\pi_l(\cdot|\mathbf{x})$. The following theorem characterizes the optimal policy that minimizes the DPO loss in Eq. (4).

**Theorem 4.3.** *Denote the policy induced by the $\boldsymbol{\theta}$ minimizing the DPO loss function in Eq.* (4) *as $\pi_{DPO}(\mathbf{y}|\mathbf{x})$. We can have*

$$\pi_{DPO}(\mathbf{y}|\mathbf{x}) \propto \left( \frac{\pi_w(\mathbf{y}|\mathbf{x})}{\pi_l(\mathbf{y}|\mathbf{x})} \right)^{\frac{1}{\beta}} \cdot \pi_{ref}(\mathbf{y}|\mathbf{x}). \tag{7}$$

The proof of Theorem 4.3 is based on taking the functional derivative of the DPO loss and the detailed proof is delayed to Appendix. Theorem 4.3 reveals that DPO modifies the reference policy $\pi_{\text{ref}}$ based on the ratio between the chosen and rejected distributions, which is generated from $\mathcal{D}_{\text{DPO}}$. DPO places more density mass where $\pi_w$ exceeds $\pi_l$. The hold of Eq. (7) does not rely on the hold of BT assumption. If the BT assumption truly holds, Eq. (7) will align with the well understood minimizer of the DPO loss (5) in certain cases.

Moreover, note that $\pi_{\text{DPO}}(\mathbf{y}|\mathbf{x})$ is also the optimal solution to the following optimization problems.

**Proposition 4.4.** *The distribution $\pi_{DPO}(\mathbf{y}|\mathbf{x})$ coincides with the solution minimizing the following loss function:*

$$\widetilde{\mathcal{L}}_{DPO}(\boldsymbol{\theta}; \mathcal{D}_{DPO}) = -\mathbb{E}_{\mathbf{x}\sim\mathcal{D}_{\mathbf{x}}, \mathbf{y}\sim\pi_{\boldsymbol{\theta}}(\cdot|\mathbf{x})}\left[\log\frac{\pi_w(\mathbf{y}|\mathbf{x})}{\pi_l(\mathbf{y}|\mathbf{x})}\right] + \beta\mathbb{D}_{KL}(\pi_{\boldsymbol{\theta}}\|\pi_{ref}) \tag{8}$$

$$= \mathbb{D}_{KL}(\pi_{\boldsymbol{\theta}}\|\pi_w) - \mathbb{D}_{KL}(\pi_{\boldsymbol{\theta}}\|\pi_l) + \beta\mathbb{D}_{KL}(\pi_{\boldsymbol{\theta}}\|\pi_{ref}). \tag{9}$$

Together with Theorem 4.3 and Proposition 4.4, there are several insights that we want to highlight.

- **DPO may deviate from RLHF.** Rafailov et al. (2023) derive the DPO objective from RLHF by the observation the language model is secretly a reward model under the BT assumption. However, when the BT assumption does not necessarily hold, Eq. (8) implies that, in a more general sense, DPO implicitly performs reward learning with a specific reward function defined by $\pi_w$ and $\pi_l$:

$$\widetilde{r}(\mathbf{y}|\mathbf{x}) = \log\left(\frac{\pi_w(\mathbf{y}|\mathbf{x})}{\pi_l(\mathbf{y}|\mathbf{x})}\right).$$

  As a result, the distribution $\mathcal{D}_{\text{DPO}}$ plays an essential role and DPO and RLHF may converge to very different policies.

- **The quality of the chosen responses matters.** Eq. (7) suggests that DPO performance is fundamentally limited by the quality of the chosen responses. More straightforwardly, when $\beta = 1$ and the rejected samples are generated from the reference model, $\pi_{\text{DPO}}(\mathbf{y}|\mathbf{x})$ is just $\pi_w(\mathbf{y}|\mathbf{x})$. It is intuitively unreasonable to expect the language model generating the responses whose quality is much better than the chosen samples after DPO. Such an intuition has also been reflected in many DPO practices. For example, Dong et al. (2024b) showcase the effectiveness of using a response improver to polish what the current model generates and use it as the chosen samples.

- **The quality of the rejected responses may not always be critical.** When $\pi_l(\mathbf{y}|\mathbf{x})$ and $\pi_w(\mathbf{y}|\mathbf{x})$ are very similar or even the same, DPO lacks a learning signal. However, the rejected distribution may not always take the fundamental role. As shown in Figure 1, imagine now we have two different distributions on the rejected samples $\pi_l'(\mathbf{y}|\mathbf{x})$ and $\pi_l''(\mathbf{y}|\mathbf{x})$ that are only different from each other on the area where $\pi_w(\mathbf{y}|\mathbf{x})$ is small. Although $\pi_l'(\mathbf{y}|\mathbf{x})$ and $\pi_l''(\mathbf{y}|\mathbf{x})$ are very different, the ratios $\pi_w(\mathbf{y}|\mathbf{x})/\pi_l'(\mathbf{y}|\mathbf{x})$ and $\pi_w(\mathbf{y}|\mathbf{x})/\pi_l''(\mathbf{y}|\mathbf{x})$ can still be similar. Thus, $\pi_l'(\mathbf{y}|\mathbf{x})$ and $\pi_l''(\mathbf{y}|\mathbf{x})$ may still lead to similar performances of DPO. In the literature, there have been some numerical results that implicitly implying such an idea. For example, Khaki et al. (2024) compare different preference data generation policies, including two based on $k$ generated answers. One is called Best-vs-worst where the chosen response is the best among the $k$ generated and the rejected response is the worst. The other one is Best-vs-random where the chosen response is again the best and the rejected response is randomly chosen from the rest $k-1$ responses. Interestingly, Khaki et al. (2024) report similar performances of Best-vs-worst and Best-vs-random across several different tasks. More numerical evidence is presented in our Section 5.

- **The role of contrastiveness between chosen and rejected samples in DPO.** Conventional wisdom in the field suggests that a larger preference gap between chosen and rejected responses enhances DPO training performance (Khaki et al., 2024; Gou and Nguyen, 2024). From Eq. (7), when $\pi_w(\mathbf{y}|\mathbf{x})$ and $\pi_w(\mathbf{y}|\mathbf{x})$ are nearly identical, DPO receives little useful signal to learn from. In this sense, greater contrastiveness helps avoid such degenerate cases. However, following what discussed above, once sufficient contrastiveness is achieved, further degrading the quality of the rejected responses may yield limited returns. A more fundamental benefit of contrastiveness appears to be that it encourages higher-quality chosen responses. This also interprets why increasing the number of candidates

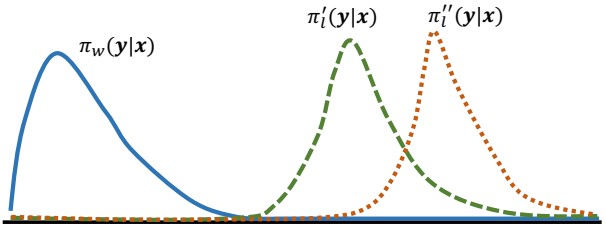

Figure 1: An illustration example on when the quality of rejected samples are not essential.

in rejection sampling tends to improve the performance: it increases the likelihood of selecting better chosen samples, while the specific choice of rejected responses is often less importantconsistent with the findings in Khaki et al. (2024).

## 4.2 Online DPO with Fixed Chosen Samples

In this subsection, we consider the setting where the chosen samples $\mathbf{y}_w$ are generated before DPO training and keep fixed. For example, $\mathbf{y}_w$ could be high-quality outputs written by human annotators or produced by a stronger, well-aligned language model. In contrast, the rejected responses $\mathbf{y}_l$ are generated from the current policy $\pi_\theta$ and are updated as training progresses. Formally, the distribution of the dataset can be written as $\bar{\mathcal{D}}_{\text{DPO}} = \mathcal{D}_\mathbf{x} \times \pi^*(\mathbf{y}_w|\mathbf{x}) \times \pi_\theta(\mathbf{y}_l|\mathbf{x})$, where $\pi^*$ denotes the fixed distribution over chosen responses. Under the distribution $\bar{\mathcal{D}}_{\text{DPO}}$, we can show that such a simplified online DPO is explicitly (almost) conducting SFT on the chosen samples. This is formalized in the following theorem.

**Theorem 4.5.** *Under the data distribution $\bar{\mathcal{D}}_{DPO}$, the derivative of the DPO loss function satisfies*

$$\nabla_{\boldsymbol{\theta}}\mathcal{L}_{DPO}(\boldsymbol{\theta}; \bar{\mathcal{D}}_{DPO}) = -\left(\frac{1}{2} + \epsilon_\beta\right)\beta \mathbb{E}_{(\mathbf{x},\mathbf{y})\sim\mathcal{D}_\mathbf{x}\times\pi^*(\cdot|\mathbf{x})}\left[\nabla_{\boldsymbol{\theta}}\log\pi_{\boldsymbol{\theta}}(\mathbf{y}|\mathbf{x})\right] + \frac{\beta^2}{4}\nabla_{\boldsymbol{\theta}}\mathbb{D}_{KL}(\pi_{\boldsymbol{\theta}}\|\pi_{ref}) + \epsilon_3,$$

$$\approx \frac{\beta}{2}\nabla_{\boldsymbol{\theta}}\left(\underbrace{-\mathbb{E}_{(\mathbf{x},\mathbf{y})\sim\mathcal{D}_\mathbf{x}\times\pi^*(\cdot|\mathbf{x})}\left[\log\pi_{\boldsymbol{\theta}}(\mathbf{y}|\mathbf{x})\right]}_{SFT\,Loss} + \frac{\beta}{2}\mathbb{D}_{KL}(\pi_{\boldsymbol{\theta}}\|\pi_{ref})\right),$$

*where $\epsilon_\beta$ represent a quantity in the order of $\beta$, $\epsilon_3$ is a three-order error, and the approximation holds when $\beta$ is chosen to be small.*

In current practice, the parameter $\beta$ is chosen between 0.03 and 0.1 under many circumstances, as seen in prior work such as Rafailov et al. (2023), as well as implementations from Cerebras.AI (Vishnevskiy, 2023) and Anyscale (Wang et al., 2024). In fact, through our dataset construction, it is also likely for $\epsilon_\beta$ to be positive. Therefore, even when $\epsilon_\beta$ is not negligibly small, the insights below can still be valid under many cases. The proof of Theorem 4.5 is based on the Taylor expansion of $\sigma(x)$, along with the fact that $\mathbf{y}_w$ and $\mathbf{y}_l$ are independently generated in $\bar{\mathcal{D}}_{\text{DPO}}$.

Theorem 4.5 establishes that the gradient of $\mathcal{L}_{\text{DPO}}(\boldsymbol{\theta}; \bar{\mathcal{D}}_{\text{DPO}})$ closely approximates the gradient of the SFT loss, with an additional regularization term penalizing the divergence between $\pi_\theta$ and $\pi_{\text{ref}}$. Intuitively, when the chosen samples are of high quality and the rejected samples are generated from the current model, DPO effectively reduces to SFT on the chosen examples. This result further reinforces our earlier observations:

- **The quality of chosen samples is critical for DPO.** Since DPO in this setting behaves like SFT on the chosen responses, the performance ceiling is determined by their quality. If the chosen samples are not of sufficiently high quality, collecting DPO data online may offer limited benefit.

- **Contrastiveness may not always be essential.** In our setup, as training progresses and the model improves, the quality of the rejected responses increases, naturally reducing the gap between chosen and rejected responses. Theorem 4.5 suggests that this reduction in contrastiveness does not significantly affect training, since the model primarily learns from the chosen samples.

# 5 Numerical Evidence

In this section, through controlled experiments and quantitative analysis, we demonstrate a strong alignment between the derived theoretical insights and the empirical findings.

## 5.1 Experiment Settings

**Base Model.** In the following numerical experiments, we are utilizing Allen-AI's open-sourced `Llama-3.1-Tulu-3-8B-SFT` checkpoint (Lambert et al., 2024) as the base model for DPO training (Sun et al., 2024). This model is exclusively supervised-finetuned (SFT) on a mix of publicly available and transparent SFT data from Meta's official pre-trained model (Dubey et al., 2024), making it possible to guarantee no data overlap during the SFT and DPO stages.

**Datasets.** According to the data recipe of our base model, we select two public datasets, LAION-AI's `Open Assistant 2` (Köpf et al., 2023) and OpenBMB's `UltraFeedback` (Cui et al., 2023), as the prompt datasets for our DPO training. We carefully curate the datasets to make sure the prompts are unique and not seen during the SFT stage.

**Data Processing.** For our experiments, we include multiple responses per prompt. For `Open Assistant 2`, we retain only first-turn dialogues and filter out prompts with fewer than 3 responses. For `UltraFeedback`, we consider two variants: the original version (ultrafeedback-original) (Cui et al., 2023), which provides 4 responses per prompt, and the tulu3 version (ultrafeedback-tulu3) (Lambert et al., 2024), which provides 2 responses per prompt. We focus on prompts that appear in both variants. After filtering, the `Open Assistant 2` and `UltraFeedback` datasets contain 4,603 and 41,633 prompts, respectively. Besides the mentioned sources of responses, to ensure the abundance of the dataset for comparison, we also leverage the responses generated by the Mistral series model (Meng et al., 2024; Jiang et al., 2023). For each completion pair, i.e., a prompt and one of its responses, we use the `Skywork-Reward-Gemma-2-27B-v0.2` (Liu et al., 2024) model as an oracle to assign quality scores. These scores serve as a proxy for data quality, enabling us to rank or categorize the samples and construct DPO datasets with different controlled qualities. We paired each prompt with five responses of varying quality, labeled as *best, high, medium, low, and worst*. We also use the filtered prompts of `UltraFeedback` dataset for on-policy response generation. For a detailed explanation, please refer to Appendix B.2.

**Evaluation.** To comprehensively assess the capabilities of our models, we employ a suite of standard evaluation benchmarks that measure diverse aspects of model performance. Based on established practices in the field, we include `AlpacaEval-2` (Dubois et al., 2024), MMLU (Hendrycks et al., 2020), `IFEval` (Zhou et al., 2023b), `TruthfulQA` (Lin et al., 2021) and GSM8K (Cobbe et al., 2021) for estimating models' abilities of general conversation, multitask understanding, instruction following, being truthful and informative, and mathematical reasoning, respectively.

For more details about the data, training and evaluation, please refer to Appendix B. Unless otherwise specified, all benchmark results reported in this work are calculated as the average of three independent runs with different random seeds, ensuring the reliability.

## 5.2 Chosen response quality dominates DPO training performance

In this part, we investigate the relative impact of chosen and rejected response quality on DPO training performance. According to the above analysis, we hypothesize that the chosen response plays a more critical role in determining the effectiveness of DPO training. To validate, we construct several DPO datasets with different qualities of the chosen and the rejected responses based on our filtered `Open Assistant 2` and `UltraFeedback` datasets. Recall that for each query, we have five responses of different qualities. Concretely, each DPO pair is synthesized under two guiding principles:

- **Fixed Chosen, Varied Rejected**: Among the multiple responses under each prompt, we lock in response of the highest quality as the chosen response, then pair it with rejected responses whose quality is systematically degraded from the relatively high to low quality.

- **Fixed Rejected, Varied Chosen**: We hold the rejected response to be at the lowest quality tier, while the chosen response is systematically varied from moderate to high quality.

As mentioned before, the quality of responses is revealed by the reward model scores. The detailed statistics can be found in Appendix B.1. To evaluate the effectiveness of DPO training, we compare the DPO-trained models with the SFT checkpoint (the untrained base model). The results are shown in Table 1.

| Dataset | Configuration | GSM8K | LC-AE2 | MMLU | IFEval | TruthfulQA |
|---|---|---|---|---|---|---|
| N/A | SFT Baseline | 76.8 | 12.7 | 62.1 | 74.3 | 46.8 |
| Open Assistant 2 **(Fixed Best)** | Best/Worst | 78.4 | 20.9 | 62.8 | 72.3 | 48.4 |
| | Best/Low | 78.6 | 19.2 | 62.6 | 72.7 | 47.4 |
| | Best/Medium | 79.3 | 19.3 | 62.8 | 71.4 | 49.1 |
| | Best/High | 78.4 | 19.6 | 62.7 | 72.1 | 47.5 |
| Open Assistant 2 **(Fixed Worst)** | Low/Worst | 77.5 | 15.2 | 61.2 | 66.5 | 47.1 |
| | Medium/Worst | 78.2 | 17.0 | 61.3 | 70.4 | 48.0 |
| | High/Worst | 78.2 | 17.4 | 61.2 | 70.2 | 47.6 |
| | Best/Worst | 78.4 | 20.9 | 62.8 | 72.3 | 48.4 |
| UltraFeedback **(Fixed Best)** | Best/Worst | 80.4 | 36.5 | 64.8 | 77.4 | 62.2 |
| | Best/Low | 80.8 | 34.5 | 63.4 | 76.9 | 58.6 |
| | Best/Medium | 80.2 | 34.2 | 63.3 | 76.7 | 59.4 |
| | Best/High | 79.0 | 33.6 | 62.5 | 76.0 | 58.7 |
| UltraFeedback **(Fixed Worst)** | Low/Worst | 79.3 | 25.8 | 61.4 | 76.5 | 56.1 |
| | Medium/Worst | 78.6 | 26.7 | 62.1 | 75.8 | 58.0 |
| | High/Worst | 79.5 | 30.9 | 63.7 | 77.0 | 61.3 |
| | Best/Worst | 80.4 | 36.5 | 64.8 | 77.4 | 62.2 |

Table 1: Results across different datasets and DPO data mixtures. Data mixture types in the "Configuration" column are formatted as **Chosen/Rejected**. Quality tiers are ranked from highest to lowest as *Best, High, Medium, Low, Worst*. "LC-AE2" is the abbreviation for Length-Controlled AlpacaEval-2 benchmark.

| Dataset | Configuration | LC-AE2 | MMLU | IFEval | TruthfulQA | GSM8K |
|---|---|---|---|---|---|---|
| N/A | SFT Baseline | 12.7 | 62.1 | 74.3 | 46.8 | 76.8 |
| Open Assistant 2 | Continual SFT | 18.7 | 60.4 | 71.5 | 46.9 | 78.7 |
| | Online-DPO | 19.0 | 60.6 | 71.8 | 47.5 | 78.6 |
| UltraFeedback | Continual SFT | 35.8 | 61.6 | 74.1 | 57.1 | 79.5 |
| | Online-DPO | 37.6 | 62.0 | 74.5 | 58.0 | 79.7 |

Table 2: Results of the chosen-fixed online DPO and continual SFT training.

Our experimental results reveal a clear asymmetric impact of chosen and rejected response quality. We find that the quality of the chosen response is the primary determinant of the model's final performance, effectively setting a knowledge ceiling. This is demonstrated by the strong, monotonic improvement in the fixed-worst setting on both datasets. As the chosen response quality increases from *Low* to *Best*, performance shows a universally climbing pattern for each of the benchmarks. This confirms that high-quality positive examples are essential for reaching high performance. Meanwhile, the role of the rejected response is more nuanced. When the chosen response is fixed to the best quality, the performance does not exhibit a monotonic trend as the quality of the rejected response increases or decreases, which indicates the quality of the rejected sample alone may not be a reliable indicator of DPO performance.

To empirically validate Theorem 4.5, we also test its central prediction: that DPO with fixed chosen responses and on-policy rejected responses approximates Supervised Fine-Tuning (SFT) on the chosen data alone. We compare two setups: (1) Online-DPO as described in Section 4.2, and (2) Continual SFT, where we perform SFT exclusively on the high-quality chosen responses from the preference set. We use the best response group mentioned in Table 1 as the training dataset. The

results are presented in Table 2. Across both datasets, the performance profiles of Online-DPO and Continual SFT are nearly identical. This striking similarity provides strong empirical support for our theory, confirming that in this setting, the DPO learning signal is overwhelmingly derived from the chosen responses, effectively reducing the process to SFT.

## 5.3 Preference gap and exposure bias might not always be essential

Building on our understanding of the impact of response quality, we now turn to investigate two additional factors frequently discussed in the context of DPO training: preference gap and exposure bias. Conventional wisdom suggests that a larger preference gap between chosen and rejected responses enhances DPO training performance (Khaki et al., 2024; Gou and Nguyen, 2024) and that exposure bias arising from on-policy data also helps improve the model's ability to learn preferences (Guo et al., 2024; Dong et al., 2024a). However, our findings respectively challenge these hypotheses, demonstrating that neither the preference gap nor exposure bias might not be as critical as previously believed. Instead, the quality of the chosen response emerges as the primary determinant of model performance, overshadowing the influence of these factors.

To investigate the relative importance of preference gap versus chosen response quality in DPO training, we create six specialized datasets derived from the ultrafeedback-original dataset through controlled modifications. The core experimental design comprises two phases:

1. We first construct four baseline datasets with two orthogonal dimensions, i.e., the preference gap size (large and small) and the chosen response quality (high and low). This yields four combinations: large gap/high-quality (LG-HQ), large gap/low-quality (LG-LQ), small gap/high-quality (SG-HQ), and small gap/low-quality (SG-LQ). Rejected responses are systematically adjusted in each pair to maintain precise gap sizes while preserving the original quality hierarchy.

2. To isolate the effect of chosen quality from gap magnitude, we introduce two additional counterfactual datasets: the first one, LG-HQ-inverse, maintains LG-HQ's high chosen quality (identical absolute scores) but reduces its gap, and the second one, SG-HQ-inverse, preserves SG-HQ's high chosen quality while expanding its gap.

With these strategically mismatched conditions, we enable direct attribution of performance variations to either gap magnitude or chosen quality dominance. We then conduct DPO training on these datasets and compare the outcome models' performance.

Our controlled experiments and their results depicted in Table 3 reveal one of our main observations: the quality of the chosen response is the dominant factor driving DPO performance, significantly outweighing the influence of the preference gap. This conclusion is supported by three key observations from our experiments. First, when controlling for the preference gap (Part 1), elevating the quality of the chosen response yields the most substantial performance improvements, delivering a +7.1 to +8.7 point gain in LC-AE2. Second, while widening the preference gap does provide a benefit, its impact is comparatively modest, contributing a smaller +3.0 to +4.6 point increase (Part 2). Finally, our counterfactual analysis (Part 3) provides the clearest evidence: when the chosen responses are identical, isolating the effect of the gap by swapping the rejected response yields only a minimal gain of +0.8 to +1.4 points. Collectively, these findings strongly suggest that DPO's effectiveness is primarily rooted in quality anchoringlearning the characteristics of the high-quality chosen responserather than in margin maximization.

To explore the impact of exposure bias, we further conduct on/off-policy data mixture experiments on the UltraFeedback dataset. We utilize the prompt datasets and rejection sampling technique to generate on-policy rejected responses and mix these on-policy responses of different quality with existing off-policy data at different ratios. We then evaluate the performance of models trained on these mixed datasets.

Our investigation of exposure bias reveals a nuanced interaction between the inclusion of policy data and the quality of the chosen response. As shown in Table 4, introducing on-policy responses yields substantial gains in LC-AE2 and GSM8K metrics. However, such benefit strictly depends on the base data quality: low-quality configurations show minimal improvement despite equivalent on-policy proportions. This confirms that exposure bias mitigation only amplifies existing quality foundations rather than compensating for low-quality chosen responses. Notably, our implemen-

| Configuration | Avg.Chs | Avg.Diff | LC-AE2 | Score | MMLU | IFEval | GSM8K |
|---|---|---|---|---|---|---|---|
| **Part 1: Effect of Chosen Quality** (Gap size is held constant) | | | | | | | |
| LG-HQ (High Quality) | -2.98 | 3.54 | 33.0 | **+8.7** | 64.9 | 76.9 | 81.2 |
| LG-LQ (Low Quality) | -5.15 | 3.45 | 24.3 | | 61.9 | 73.8 | 80.5 |
| SG-HQ (High Quality) | -3.56 | 1.34 | 28.4 | **+7.1** | 64.0 | 74.2 | 80.8 |
| SG-LQ (Low Quality) | -6.59 | 1.49 | 21.3 | | 62.6 | 72.3 | 78.0 |
| **Part 2: Effect of Preference Gap** (Chosen quality is held constant) | | | | | | | |
| LG-HQ (Large Gap) | -2.98 | 3.54 | 33.0 | **+4.6** | 64.9 | 76.9 | 81.2 |
| SG-HQ (Small Gap) | -3.56 | 1.34 | 28.4 | | 64.0 | 74.2 | 80.8 |
| LG-LQ (Large Gap) | -5.15 | 3.45 | 24.3 | **+3.0** | 61.9 | 73.8 | 80.5 |
| SG-LQ (Small Gap) | -6.59 | 1.49 | 21.3 | | 62.6 | 72.3 | 78.0 |
| **Part 3: Effect of Gap via Counterfactuals** (Chosen quality is identical) | | | | | | | |
| LG-HQ (Large Gap) | -2.98 | 3.54 | 33.0 | **+1.4** | 64.9 | 76.9 | 81.2 |
| LG-HQ-inv (Small Gap) | -2.98 | 1.92 | 31.6 | | 64.7 | 76.5 | 81.3 |
| SG-HQ-inv (Large Gap) | -3.56 | 4.51 | 29.2 | **+0.8** | 64.2 | 75.1 | 80.5 |
| SG-HQ (Small Gap) | -3.56 | 1.34 | 28.4 | | 64.0 | 74.2 | 80.8 |

Table 3: Disentangling the effects of chosen quality and preference gap. This analysis compares the performance delta (by `LC-AE2`) from improving quality versus widening the gap. The gain from higher chosen quality (**Part 1: +7.1 to +8.7 points**) is consistently and significantly larger than the gain from a wider preference gap (**Part 2: +3.0 to +4.6 points**). The counterfactuals in **Part 3** further confirm this, showing that altering the gap while keeping quality constant has only a minor effect (**+0.8 to +1.4 points**).

| Avg.Chs | On-Pol.% | LC-AE2 | MMLU | IFEval | TruthfulQA | GSM8K |
|---|---|---|---|---|---|---|
| $-0.93$ | 0 | 34.5 | 63.4 | 76.9 | 58.6 | 80.8 |
| $-4.18$ | 0 | 25.8 | 61.4 | 76.5 | 56.1 | 79.3 |
| $-0.93$ | 10% | 39.4 | 63.2 | 76.7 | 56.7 | 82.2 |
| $-0.93$ | 20% | 39.2 | 63.4 | 76.2 | 56.4 | 81.9 |
| $-4.18$ | 10% | 27.7 | 61.3 | 76.3 | 56.2 | 80.0 |
| $-4.18$ | 20% | 27.4 | 61.5 | 76.5 | 56.2 | 79.6 |

Table 4: Results across data mixtures of different on-policy data ratios. The "On-Pol.%" stands for on-policy data ratio in percentage.

tation adopts the commonly used on-policy data ratios in the literature, as excessive reliance on such data is very likely to induce training instability that can lead to a significant drop in model performance (Lambert et al., 2024; Deng et al., 2025).

# 6 Conclusion

This work provides a theoretical and empirical analysis of the role of preference data in DPO. We demonstrate that the quality of chosen responses is the primary driver of DPO performance, whereas the quality of rejected responses plays a comparatively less important role. Our results also clarify the mechanism behind commonly used practices such as increasing contrastiveness, showing that their effectiveness stems largely from improving the quality of chosen responses. Our empirical studies across multiple tasks confirm these insights, highlighting that improving the absolute quality of chosen responses consistently yields better outcomes. These findings provide practical guidance for building preference datasets and raise important considerations for future alignment strategies, including better data selection, more targeted annotation protocols, and extensions to more complex preference structures.

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

# A  Technical Details

## A.1  Proof to Theorem 4.1

For brevity, let us denote the relative logit of $\pi_\theta$ and $\pi_{ref}$ by $f_\theta(x,y) = \log \frac{\pi_\theta(y|x)}{\pi_{ref}(y|x)}$. Then

$$\mathcal{L}(\theta,\mathcal{D}) = \mathbb{E}_{(x,y_w,y_l)\sim\mathcal{D}}[-\log(\sigma(\beta(f_\theta(x,y_w) - f_\theta(x,y_l))))].$$

With the gradient update

$$\theta_{t+1} = \theta_t - \alpha \cdot \nabla_\theta \mathcal{L}(\theta_t, \mathcal{D}), \tag{10}$$

the two policies $\pi_{\theta_{t+1}}(y|x)$ and $\pi_{\theta_t}(y|x)$ have the following relationship

$$f_{\theta_{t+1}}(x,y) = f_{\theta_t}(x,y) - \alpha g_{\theta_t}(x,y) + O(\alpha^2)$$

via second-order approximation where $g_\theta(x,y)$ denotes the functional derivative of $\mathcal{L}$ with respect to $f_\theta$, that is, $g_\theta(x,y) = \frac{\delta \mathcal{L}}{\delta f_\theta}(x,y)$. Note that $\alpha$ is usually taken around $10^{-5}$, so $O(\alpha^2)$ is negligible. In order to compute $g_\theta$, we explicitly express the dependency of $\mathcal{L}$ to $f_\theta$ as $\mathcal{L}[f_\theta]$. Since

$$\mathcal{L}[f + \epsilon\tilde{f}] = \mathbb{E}_{(x,y_w,y_l)\sim\mathcal{D}}\left[-\log\sigma\left(\beta(f(x,y_w) + \epsilon\tilde{f}(x,y_w) - f(x,y_l) - \epsilon\tilde{f}(x,y_l))\right)\right],$$

for a test function $\tilde{f}$, the functional differential is

$$\delta\mathcal{L}[f,\tilde{f}] = \left.\frac{\partial}{\partial\epsilon}\mathcal{L}[f + \epsilon\tilde{f}]\right|_{\epsilon=0}$$

$$= -\beta\mathbb{E}_{\mathcal{D}}\left[\{1 - \sigma\left(\beta(f(x,y_w) - f(x,y_l))\right)\}\left(\tilde{f}(x,y_w) - \tilde{f}(x,y_l)\right)\right]. \tag{11}$$

Suppose $\mathbb{E}_{\mathcal{D}}$ is taken over population. Let us define $q(y_1,y_2|x) = \frac{p_{Y_1,Y_2|X}(y_1,y_2|x)+p_{Y_1,Y_2|X}(y_2,y_1|x)}{2}$. Then (11) extends to

$$\delta\mathcal{L}[f,\tilde{f}] = -\beta\iiint \frac{e^{-\beta(f_\theta(x,y_1)-f_\theta(x,y_2))}}{1 + e^{-\beta(f_\theta(x,y_1)-f_\theta(x,y_2))}}(\tilde{f}(x,y_1) - \tilde{f}(x,y_2))p_X(x)\left(p_{Y_1,Y_2|X}(y_1,y_2|x) + p_{Y_1,Y_2|X}(y_2,y_1|x)\right)$$

$$\cdot \mathbb{P}(y_1 \succ y_2|x)dy_2 dy_1 dx$$

$$= \iint -\beta\int \frac{e^{-\beta(f_\theta(x,y_1)-f_\theta(x,y_2))}}{1 + e^{-\beta(f_\theta(x,y_1)-f_\theta(x,y_2))}}p_X(x)2q(y_1,y_2|x)\mathbb{P}(y_1 \succ y_2|x)dy_2\tilde{f}(x,y_1)dy_1 dx$$

$$+ \iint \beta\int \frac{e^{-\beta(f_\theta(x,y_1)-f_\theta(x,y_2))}}{1 + e^{-\beta(f_\theta(x,y_1)-f_\theta(x,y_2))}}p_X(x)2q(y_1,y_2|x)\mathbb{P}(y_1 \succ y_2|x)dy_1\tilde{f}(x,y_2)dy_2 dx.$$

As the functional derivative $\frac{\delta\mathcal{L}}{\delta f}$ is defined by equation $\delta\mathcal{L}[f,\tilde{f}] = \iint \frac{\delta\mathcal{L}}{\delta f}(x,y)\tilde{f}(x,y)dydx$, it follows that

$$\frac{\delta\mathcal{L}}{\delta f_\theta}(x,y) = -\beta\int \frac{e^{-\beta(f_\theta(x,y)-f_\theta(x,y_2))}}{1 + e^{-\beta(f_\theta(x,y)-f_\theta(x,y_2))}}p_X(x)2q(y,y_2|x)\mathbb{P}(y \succ y_2|x)dy_2$$

$$+ \beta\int \frac{e^{-\beta(f_\theta(x,y_1)-f_\theta(x,y))}}{1 + e^{-\beta(f_\theta(x,y_1)-f_\theta(x,y))}}p_X(x)2q(y_1,y|x)\mathbb{P}(y \prec y_1|x)dy_1$$

$$= -\beta\int \left(1 - \frac{e^{-\beta(f_\theta(x,y_2)-f_\theta(x,y))}}{1 + e^{-\beta(f_\theta(x,y_2)-f_\theta(x,y))}}\right)p_X(x)2q(y,y_2|x)\mathbb{P}(y \succ y_2|x)dy_2$$

$$+ \beta\int \frac{e^{-\beta(f_\theta(x,y_1)-f_\theta(x,y))}}{1 + e^{-\beta(f_\theta(x,y_1)-f_\theta(x,y))}}p_X(x)2q(y,y_1|x)\mathbb{P}(y \prec y_1|x)dy_1$$

$$= -\beta\int p_X(x)2q(y,y_2|x)\mathbb{P}(y \succ y_2|x)dy_2 + \beta\int \frac{e^{-\beta(f_\theta(x,y_2)-f_\theta(x,y))}}{1 + e^{-\beta(f_\theta(x,y_2)-f_\theta(x,y))}}p_X(x)2q(y,y_2|x)\mathbb{P}(y \succ y_2|x)dy_2$$

$$+ \beta\int \frac{e^{-\beta(f_\theta(x,y_2)-f_\theta(x,y))}}{1 + e^{-\beta(f_\theta(x,y_2)-f_\theta(x,y))}}p_X(x)2q(y,y_2|x)\mathbb{P}(y \prec y_2|x)dy_2 \tag{12}$$

$$= -\beta\int p_X(x)2q(y,y_2|x)\mathbb{P}(y \succ y_2|x)dy_2 + \beta\int \frac{e^{-\beta(f_\theta(x,y_2)-f_\theta(x,y))}}{1 + e^{-\beta(f_\theta(x,y_2)-f_\theta(x,y))}}p_X(x)2q(y,y_2|x)dy_2. \tag{13}$$

In (12), dummy variable $y_1$ in the last integral is substituted by $y_2$. (13) uses $\mathbb{P}(y \succ y_2|x) + \mathbb{P}(y \prec y_2|x) = 1$.

$\frac{\delta \mathcal{L}}{\delta f_\theta}$ can be further reduced by realizing that (13) is an expectation with respect to $y_2$ over a conditional distribution as follows. (13) is integrated by $y_2$ with the term $p_X(x)q(y,y_2|x)$. If $(X, Y_1, Y_2) \sim \mathcal{D}_u$, then this is the joint density. However, $x$ and $y$ are arguments of $\frac{\delta \mathcal{L}}{\delta f_\theta}$ and only $y_2$ is being integrated. In order to simplify the integrals, we define $\mathcal{D}_u|(X, Y_1) = (x, y)$ as the conditional distribution of $Y_2$ given $X$ and $Y_1$ under $\mathcal{D}_u$. Equivalently, the density of $Y_2 \sim \mathcal{D}_u|(X, Y_1) = (x, y)$ is $\frac{p_X(x)q(y,y_2|x)}{p_{X,Y_1}(x,y)}$, where $p_{X,Y_1}(x,y) = \int p_X(x)q(y,y_2|x)dy_2$ is the marginal distribution of $(X, Y_1)$ on $\mathcal{D}_u$. Intuitively, it amounts to first sampling $(X, Y_1, Y_2) \sim \mathcal{D}_u$ and then considering only the case where $(X, Y_1) = (x, y)$. With this definition, the first integral in (13) is

$$\int p_X(x)2q(y,y_2|x)\mathbb{P}(y \succ y_2|x)dy_2 = \mathbb{E}_{Y_2 \sim \mathcal{D}_u|(X,Y_1)=(x,y)}[\mathbb{P}(y \sim Y_2|x)]. \qquad (14)$$

The second integral in (13) is reduced using the same distribution. Recall that the BT model with a reward $r$ is defined as

$$\mathbb{P}^{BT}(y_1 \succ y_2|x) = \frac{\exp(r(x,y_1))}{\exp(r(x,y_1)) + \exp(r(x,y_2))} = \sigma(r(x,y_1) - r(x,y_2)).$$

Let us denote $\mathbb{P}_\theta^{BT}$ the BT model with reward $\beta f_\theta$, that is,

$$\mathbb{P}_\theta^{BT}(y_1 \succ y_2|x) = \sigma(\beta(f_\theta(x,y_1) - f_\theta(x,y_2))).$$

Then

$$\int \frac{e^{-\beta(f_\theta(x,y_2)-f_\theta(x,y))}}{1 + e^{-\beta(f_\theta(x,y_2)-f_\theta(x,y))}} p_X(x)2q(y,y_2|x)dy_2 = \int \sigma(\beta(f_\theta(x,y) - f_\theta(x,y_2)))p_X(x)2q(y,y_2|x)dy_2$$

$$= \int \mathbb{P}_\theta^{BT}(y \succ y_2|x)p_X(x)2q(y,y_2|x)dy_2$$

$$= \mathbb{E}_{Y_2 \sim \mathcal{D}_u|(X,Y_1)=(x,y)}[\mathbb{P}_\theta^{BT}(y \succ Y_2|x)|x,y]p_{X,Y_1}(x,y). \qquad (15)$$

Plugging (14) and (15) in (13) leads to

$$g_\theta(x,y) = -\beta \mathbb{E}_{Y_2 \sim \mathcal{D}_u|X,Y_1} \left[ \mathbb{P}(y \succ Y_2|x) - \mathbb{P}_\theta^{BT}(y \succ Y_2|x)|x,y \right] 2p_{X,Y_1}(x,y). \qquad (16)$$

## A.2 Proof to Theorem 4.3

For simplicity, let us consider a fixed context $\mathbf{x}$, and we take the functional derivative of the DPO loss in Eq. (4) with respect to $\pi_\theta(\mathbf{y}|\mathbf{x})$,

$$\frac{\partial \mathcal{L}_{\text{DPO}}(\boldsymbol{\theta}; \mathcal{D}_{\text{DPO}})}{\partial \pi_\theta(\mathbf{y}|\mathbf{x})} = \mathbb{E}_{(\mathbf{y}_w, \mathbf{y}_l) \sim \mathcal{D}_{\text{DPO}}} \left[ \frac{\partial \log \sigma \left( \beta \log \frac{\pi_\theta(\mathbf{y}_w|\mathbf{x})}{\pi_{\text{ref}}(\mathbf{y}_w|\mathbf{x})} - \beta \log \frac{\pi_\theta(\mathbf{y}_l|\mathbf{x})}{\pi_{\text{ref}}(\mathbf{y}_l|\mathbf{x})} \right)}{\partial \pi_\theta(\mathbf{y}|\mathbf{x})} \right]. \qquad (17)$$

Note that the derivates are only non-zero either when $\mathbf{y} = \mathbf{y}_w$ or $\mathbf{y} = \mathbf{y}_l$. Therefore, when $\mathbf{y} = \mathbf{y}_w$,

$$\frac{\partial \log \sigma \left( \beta \log \frac{\pi_\theta(\mathbf{y}|\mathbf{x})}{\pi_{\text{ref}}(\mathbf{y}|\mathbf{x})} - \beta \log \frac{\pi_\theta(\mathbf{y}_l|\mathbf{x})}{\pi_{\text{ref}}(\mathbf{y}_l|\mathbf{x})} \right)}{\partial \pi_\theta(\mathbf{y}|\mathbf{x})} \delta_{\mathbf{y}=\mathbf{y}_w}$$

$$= \left( 1 - \sigma \left( \beta \log \frac{\pi_\theta(\mathbf{y}|\mathbf{x})}{\pi_{\text{ref}}(\mathbf{y}|\mathbf{x})} - \beta \log \frac{\pi_\theta(\mathbf{y}_l|\mathbf{x})}{\pi_{\text{ref}}(\mathbf{y}_l|\mathbf{x})} \right) \right) \frac{\delta_{\mathbf{y}=\mathbf{y}_w}}{\pi_\theta(\mathbf{y}|\mathbf{x})}, \qquad (18)$$

where $\delta$ is the Kronecker delta, and the equality holds since $\partial(\log \sigma(z))/\partial z = 1 - \sigma(z)$. Similarly, we can have that when $\mathbf{y} = \mathbf{y}_l$,

$$\frac{\partial \log \sigma \left( \beta \log \frac{\pi_\theta(\mathbf{y}_w|\mathbf{x})}{\pi_{\text{ref}}(\mathbf{y}_w|\mathbf{x})} - \beta \log \frac{\pi_\theta(\mathbf{y}|\mathbf{x})}{\pi_{\text{ref}}(\mathbf{y}|\mathbf{x})} \right)}{\partial \pi_\theta(\mathbf{y}|\mathbf{x})} \delta_{\mathbf{y}=\mathbf{y}_l}$$

$$= -\left( 1 - \sigma \left( \beta \log \frac{\pi_\theta(\mathbf{y}_w|\mathbf{x})}{\pi_{\text{ref}}(\mathbf{y}_w|\mathbf{x})} - \beta \log \frac{\pi_\theta(\mathbf{y}|\mathbf{x})}{\pi_{\text{ref}}(\mathbf{y}|\mathbf{x})} \right) \right) \frac{\delta_{\mathbf{y}=\mathbf{y}_l}}{\pi_\theta(\mathbf{y}|\mathbf{x})}. \qquad (19)$$

By plugging Eqs. (18) and (19) into Eq. (17), we can get

$$
\frac{\partial \mathcal{L}_{\text{DPO}}(\boldsymbol{\theta}; \mathcal{D}_{\text{DPO}})}{\partial \pi_{\boldsymbol{\theta}}(\mathbf{y}|\mathbf{x})}
$$

$$
= \frac{\pi_w(\mathbf{y}|\mathbf{x})}{\pi_{\boldsymbol{\theta}}(\mathbf{y}|\mathbf{x})} \mathbb{E}_{\mathbf{y}_l \sim \pi_l(\cdot|\mathbf{x})} \left[ \sigma \left( \beta \log \frac{\pi_{\boldsymbol{\theta}}(\mathbf{y}_l|\mathbf{x})}{\pi_{\text{ref}}(\mathbf{y}_l|\mathbf{x})} - \beta \log \frac{\pi_{\boldsymbol{\theta}}(\mathbf{y}|\mathbf{x})}{\pi_{\text{ref}}(\mathbf{y}|\mathbf{x})} \right) \right]
$$

$$
- \frac{\pi_l(\mathbf{y}|\mathbf{x})}{\pi_{\boldsymbol{\theta}}(\mathbf{y}|\mathbf{x})} \mathbb{E}_{\mathbf{y}_w \sim \pi_w(\cdot|\mathbf{x})} \left[ \sigma \left( \beta \log \frac{\pi_{\boldsymbol{\theta}}(\mathbf{y}|\mathbf{x})}{\pi_{\text{ref}}(\mathbf{y}|\mathbf{x})} - \beta \log \frac{\pi_{\boldsymbol{\theta}}(\mathbf{y}_w|\mathbf{x})}{\pi_{\text{ref}}(\mathbf{y}_w|\mathbf{x})} \right) \right]
$$

$$
= \frac{\pi_w(\mathbf{y}|\mathbf{x})}{\pi_{\boldsymbol{\theta}}(\mathbf{y}|\mathbf{x})} \mathbb{E}_{\mathbf{y}_l \sim \pi_l(\cdot|\mathbf{x})} \left[ \sigma \left( \beta \log \frac{\pi_{\boldsymbol{\theta}}(\mathbf{y}_l|\mathbf{x})}{\pi_{\text{ref}}(\mathbf{y}_l|\mathbf{x})} - \beta \log \frac{\pi_{\boldsymbol{\theta}}(\mathbf{y}|\mathbf{x})}{\pi_{\text{ref}}(\mathbf{y}|\mathbf{x})} \right) \right]
$$

$$
- \frac{\pi_w(\mathbf{y}|\mathbf{x})}{\pi_{\boldsymbol{\theta}}(\mathbf{y}|\mathbf{x})} \mathbb{E}_{\mathbf{y}_l \sim \pi_l(\cdot|\mathbf{x})} \left[ \frac{\pi_l(\mathbf{y}|\mathbf{x})}{\pi_w(\mathbf{y}|\mathbf{x})} \frac{\pi_w(\mathbf{y}_l|\mathbf{x})}{\pi_l(\mathbf{y}_l|\mathbf{x})} \sigma \left( \beta \log \frac{\pi_{\boldsymbol{\theta}}(\mathbf{y}|\mathbf{x})}{\pi_{\text{ref}}(\mathbf{y}|\mathbf{x})} - \beta \log \frac{\pi_{\boldsymbol{\theta}}(\mathbf{y}_l|\mathbf{x})}{\pi_{\text{ref}}(\mathbf{y}_l|\mathbf{x})} \right) \right]
$$

$$
= \frac{\pi_w(\mathbf{y}|\mathbf{x})}{\pi_{\boldsymbol{\theta}}(\mathbf{y}|\mathbf{x})} \mathbb{E}_{\mathbf{y}_l \sim \pi_l(\cdot|\mathbf{x})} \left[ \sigma \left( \beta \log \frac{\pi_{\boldsymbol{\theta}}(\mathbf{y}_l|\mathbf{x})}{\pi_{\text{ref}}(\mathbf{y}_l|\mathbf{x})} - \beta \log \frac{\pi_{\boldsymbol{\theta}}(\mathbf{y}|\mathbf{x})}{\pi_{\text{ref}}(\mathbf{y}|\mathbf{x})} \right) \right]
$$

$$
- \frac{\pi_w(\mathbf{y}|\mathbf{x})}{\pi_{\boldsymbol{\theta}}(\mathbf{y}|\mathbf{x})} \mathbb{E}_{\mathbf{y}_l \sim \pi_l(\cdot|\mathbf{x})} \left[ \frac{\pi_l(\mathbf{y}|\mathbf{x})}{\pi_w(\mathbf{y}|\mathbf{x})} \frac{\pi_w(\mathbf{y}_l|\mathbf{x})}{\pi_l(\mathbf{y}_l|\mathbf{x})} e^{\beta \log \frac{\pi_{\boldsymbol{\theta}}(\mathbf{y}|\mathbf{x})}{\pi_{\text{ref}}(\mathbf{y}|\mathbf{x})} - \beta \log \frac{\pi_{\boldsymbol{\theta}}(\mathbf{y}_l|\mathbf{x})}{\pi_{\text{ref}}(\mathbf{y}_l|\mathbf{x})}} \sigma \left( \beta \log \frac{\pi_{\boldsymbol{\theta}}(\mathbf{y}_l|\mathbf{x})}{\pi_{\text{ref}}(\mathbf{y}_l|\mathbf{x})} - \beta \log \frac{\pi_{\boldsymbol{\theta}}(\mathbf{y}|\mathbf{x})}{\pi_{\text{ref}}(\mathbf{y}|\mathbf{x})} \right) \right]
$$

$$
= \frac{\pi_w(\mathbf{y}|\mathbf{x})}{\pi_{\boldsymbol{\theta}}(\mathbf{y}|\mathbf{x})} \mathbb{E}_{\mathbf{y}_l \sim \pi_l(\cdot|\mathbf{x})} \left[ \sigma \left( \beta \log \frac{\pi_{\boldsymbol{\theta}}(\mathbf{y}_l|\mathbf{x})}{\pi_{\text{ref}}(\mathbf{y}_l|\mathbf{x})} - \beta \log \frac{\pi_{\boldsymbol{\theta}}(\mathbf{y}|\mathbf{x})}{\pi_{\text{ref}}(\mathbf{y}|\mathbf{x})} \right) \right]
$$

$$
- \frac{\pi_w(\mathbf{y}|\mathbf{x})}{\pi_{\boldsymbol{\theta}}(\mathbf{y}|\mathbf{x})} \mathbb{E}_{\mathbf{y}_l \sim \pi_l(\cdot|\mathbf{x})} \left[ e^{- \log \frac{\pi_w(\mathbf{y}|\mathbf{x})}{\pi_l(\mathbf{y}|\mathbf{x})} + \log \frac{\pi_l(\mathbf{y}_l|\mathbf{x})}{\pi_w(\mathbf{y}_l|\mathbf{x})} + \beta \log \frac{\pi_{\boldsymbol{\theta}}(\mathbf{y}|\mathbf{x})}{\pi_{\text{ref}}(\mathbf{y}|\mathbf{x})} - \beta \log \frac{\pi_{\boldsymbol{\theta}}(\mathbf{y}_l|\mathbf{x})}{\pi_{\text{ref}}(\mathbf{y}_l|\mathbf{x})}} \sigma \left( \beta \log \frac{\pi_{\boldsymbol{\theta}}(\mathbf{y}_l|\mathbf{x})}{\pi_{\text{ref}}(\mathbf{y}_l|\mathbf{x})} - \beta \log \frac{\pi_{\boldsymbol{\theta}}(\mathbf{y}|\mathbf{x})}{\pi_{\text{ref}}(\mathbf{y}|\mathbf{x})} \right) \right],
$$

where the second equality is by changing the probability measure, and the third equality holds due to the fact that $\sigma(-x) = \sigma(x)e^{-x}$. Therefore, by setting $\pi_{\boldsymbol{\theta}}(\mathbf{y}|\mathbf{x})$ for every $\mathbf{y}$ as,

$$
\pi_{\boldsymbol{\theta}}(\mathbf{y}|\mathbf{x}) \propto \pi_{\text{ref}}(\mathbf{y}|\mathbf{x}) \left( \frac{\pi_w(\mathbf{y}|\mathbf{x})}{\pi_l(\mathbf{y}|\mathbf{x})} \right)^{\frac{1}{\beta}},
$$

the functional derivate $\frac{\partial \mathcal{L}_{\text{DPO}}(\boldsymbol{\theta}; \mathcal{D}_{\text{DPO}})}{\partial \pi_{\boldsymbol{\theta}}(\mathbf{y}|\mathbf{x})}$ is always 0, since $- \log \frac{\pi_w(\mathbf{y}|\mathbf{x})}{\pi_l(\mathbf{y}|\mathbf{x})} + \log \frac{\pi_l(\mathbf{y}_l|\mathbf{x})}{\pi_w(\mathbf{y}_l|\mathbf{x})} + \beta \log \frac{\pi_{\boldsymbol{\theta}}(\mathbf{y}|\mathbf{x})}{\pi_{\text{ref}}(\mathbf{y}|\mathbf{x})} - \beta \log \frac{\pi_{\boldsymbol{\theta}}(\mathbf{y}_l|\mathbf{x})}{\pi_{\text{ref}}(\mathbf{y}_l|\mathbf{x})}$ is going to be equal to 0. We finish the proof.

### A.3 Proof to Theorem 4.5

First, apply Taylor Expansion to $\sigma(z) := 1/(1 + e^{-z})$,

$$
\sigma(z) = \frac{1}{2} + \frac{1}{4}z + o(z^3),
$$

since the second derivate of $\sigma(z)$ at $z = 0$ is equal to 0. Another useful fact that we will heavily rely on is that

$$
\mathbb{E}_{(\mathbf{x}, \mathbf{y}_w, \mathbf{y}_l) \sim \bar{\mathcal{D}}_{\text{DPO}}}[\nabla_{\boldsymbol{\theta}} \log \pi_{\boldsymbol{\theta}}(\mathbf{y}_l|\mathbf{x})] = \mathbb{E}_{(\mathbf{x}, \mathbf{y}_l) \sim \mathcal{D}_{\mathbf{x}} \times \pi_{\boldsymbol{\theta}}(\cdot|\mathbf{x})}[\nabla_{\boldsymbol{\theta}} \log \pi_{\boldsymbol{\theta}}(\mathbf{y}_l|\mathbf{x})]
$$

$$
= \mathbb{E}_{\mathbf{x} \sim \mathcal{D}_{\mathbf{x}}} \left[ \int \pi_{\boldsymbol{\theta}}(\mathbf{y}_l|\mathbf{x}) \nabla_{\boldsymbol{\theta}} \log \pi_{\boldsymbol{\theta}}(\mathbf{y}_l|\mathbf{x}) d\mathbf{y}_l \right]
$$

$$
= \mathbb{E}_{\mathbf{x} \sim \mathcal{D}_{\mathbf{x}}} \left[ \int \nabla_{\boldsymbol{\theta}} \pi_{\boldsymbol{\theta}}(\mathbf{y}_l|\mathbf{x}) d\mathbf{y}_l \right]
$$

$$
= \mathbb{E}_{\mathbf{x} \sim \mathcal{D}_{\mathbf{x}}} \left[ \nabla_{\boldsymbol{\theta}} \int \pi_{\boldsymbol{\theta}}(\mathbf{y}_l|\mathbf{x}) d\mathbf{y}_l \right]
$$

$$
= 0, \tag{20}
$$

where the first equality holds due to how we construct $\bar{\mathcal{D}}_{\text{DPO}}$ and the last inequality is because of $\int \pi_{\boldsymbol{\theta}}(\mathbf{y}_l|\mathbf{x}) d\mathbf{y}_l = 1$.

From Eq. (6), we can have

$$\nabla_{\boldsymbol{\theta}}\mathcal{L}_{\text{DPO}}(\boldsymbol{\theta};\bar{\mathcal{D}}_{\text{DPO}})$$

$$= -\beta\mathbb{E}_{(\mathbf{x},\mathbf{y}_w,\mathbf{y}_l)\sim\bar{\mathcal{D}}_{\text{DPO}}}\left[\sigma(\hat{r}_{\boldsymbol{\theta}}(\mathbf{x},\mathbf{y}_l) - \hat{r}_{\boldsymbol{\theta}}(\mathbf{x},\mathbf{y}_w))\left[\nabla_{\boldsymbol{\theta}}\log\pi_{\boldsymbol{\theta}}(\mathbf{y}_w|\mathbf{x}) - \nabla_{\boldsymbol{\theta}}\log\pi_{\boldsymbol{\theta}}(\mathbf{y}_l|\mathbf{x})\right]\right]$$

$$\approx -\beta\mathbb{E}_{(\mathbf{x},\mathbf{y}_w,\mathbf{y}_l)\sim\bar{\mathcal{D}}_{\text{DPO}}}\left[\left(\frac{1}{2} + \frac{1}{4}(\hat{r}_{\boldsymbol{\theta}}(\mathbf{x},\mathbf{y}_l) - \hat{r}_{\boldsymbol{\theta}}(\mathbf{x},\mathbf{y}_w))\right)\left[\nabla_{\boldsymbol{\theta}}\log\pi_{\boldsymbol{\theta}}(\mathbf{y}_w|\mathbf{x}) - \nabla_{\boldsymbol{\theta}}\log\pi_{\boldsymbol{\theta}}(\mathbf{y}_l|\mathbf{x})\right]\right]$$

$$= -\frac{\beta}{2}\mathbb{E}_{(\mathbf{x},\mathbf{y}_w,\mathbf{y}_l)\sim\bar{\mathcal{D}}_{\text{DPO}}}\left[\nabla_{\boldsymbol{\theta}}\log\pi_{\boldsymbol{\theta}}(\mathbf{y}_w|\mathbf{x})\right]$$
$$- \frac{\beta}{4}\mathbb{E}_{(\mathbf{x},\mathbf{y}_w,\mathbf{y}_l)\sim\bar{\mathcal{D}}_{\text{DPO}}}\left[(\hat{r}_{\boldsymbol{\theta}}(\mathbf{x},\mathbf{y}_l) - \hat{r}_{\boldsymbol{\theta}}(\mathbf{x},\mathbf{y}_w))\right]\left[\nabla_{\boldsymbol{\theta}}\log\pi_{\boldsymbol{\theta}}(\mathbf{y}_w|\mathbf{x}) - \nabla_{\boldsymbol{\theta}}\log\pi_{\boldsymbol{\theta}}(\mathbf{y}_l|\mathbf{x})\right]$$

$$= -\frac{\beta}{2}\mathbb{E}_{\mathbb{E}_{(\mathbf{x},\mathbf{y})\sim\mathcal{D}_{\mathbf{x}}\times\pi^*(\cdot|\mathbf{x})}}\left[\nabla_{\boldsymbol{\theta}}\log\pi_{\boldsymbol{\theta}}(\mathbf{y}|\mathbf{x})\right] - \frac{\beta}{4}\mathbb{E}_{(\mathbf{x},\mathbf{y}_w,\mathbf{y}_l)\sim\bar{\mathcal{D}}_{\text{DPO}}}\left[(\hat{r}_{\boldsymbol{\theta}}(\mathbf{x},\mathbf{y}_l) - \hat{r}_{\boldsymbol{\theta}}(\mathbf{x},\mathbf{y}_w))\nabla_{\boldsymbol{\theta}}\log\pi_{\boldsymbol{\theta}}(\mathbf{y}_w|\mathbf{x})\right]$$
$$+ \frac{\beta}{4}\mathbb{E}_{(\mathbf{x},\mathbf{y}_w,\mathbf{y}_l)\sim\bar{\mathcal{D}}_{\text{DPO}}}\left[\hat{r}_{\boldsymbol{\theta}}(\mathbf{x},\mathbf{y}_l)\nabla_{\boldsymbol{\theta}}\log\pi_{\boldsymbol{\theta}}(\mathbf{y}_l|\mathbf{x})\right]$$

$$= -\left(\frac{1}{2} + \epsilon_{\beta}\right)\beta\mathbb{E}_{\mathbb{E}_{(\mathbf{x},\mathbf{y})\sim\mathcal{D}_{\mathbf{x}}\times\pi^*(\cdot|\mathbf{x})}}\left[\nabla_{\boldsymbol{\theta}}\log\pi_{\boldsymbol{\theta}}(\mathbf{y}|\mathbf{x})\right] + \frac{\beta^2}{4}\nabla_{\boldsymbol{\theta}}\mathbb{D}_{\text{KL}}(\pi_{\boldsymbol{\theta}}\|\pi_{\text{ref}}),$$

where the second and the third equations are based on the repeated use of Eq. (20), and the last equality holds because

$$\nabla_{\boldsymbol{\theta}}\mathbb{D}_{\text{KL}}(\pi_{\boldsymbol{\theta}}\|\pi_{\text{ref}}) = \nabla_{\boldsymbol{\theta}}\mathbb{E}_{(\mathbf{x},\mathbf{y}_l)\sim\mathcal{D}_{\mathbf{x}}\times\pi_{\boldsymbol{\theta}}(\cdot|\mathbf{x})}\left[\log\frac{\pi_{\boldsymbol{\theta}}(\mathbf{y}_l|\mathbf{x})}{\pi_{\text{ref}}(\mathbf{y}_l|\mathbf{x})}\right]$$

$$= \mathbb{E}_{\mathbf{x}\sim\mathcal{D}_{\mathbf{x}}}\left[\int \pi_{\boldsymbol{\theta}}(\mathbf{y}_l|\mathbf{x})\log\frac{\pi_{\boldsymbol{\theta}}(\mathbf{y}_l|\mathbf{x})}{\pi_{\text{ref}}(\mathbf{y}_l|\mathbf{x})}d\mathbf{y}_l\right]$$

$$= \mathbb{E}_{\mathbf{x}\sim\mathcal{D}_{\mathbf{x}}}\left[\int \nabla_{\boldsymbol{\theta}}\left(\pi_{\boldsymbol{\theta}}(\mathbf{y}_l|\mathbf{x})\log\frac{\pi_{\boldsymbol{\theta}}(\mathbf{y}_l|\mathbf{x})}{\pi_{\text{ref}}(\mathbf{y}_l|\mathbf{x})}\right)d\mathbf{y}_l\right]$$

$$= \mathbb{E}_{\mathbf{x}\sim\mathcal{D}_{\mathbf{x}}}\left[\int \log\frac{\pi_{\boldsymbol{\theta}}(\mathbf{y}_l|\mathbf{x})}{\pi_{\text{ref}}(\mathbf{y}_l|\mathbf{x})}\nabla_{\boldsymbol{\theta}}\pi_{\boldsymbol{\theta}}(\mathbf{y}_l|\mathbf{x})d\mathbf{y}_l\right] + \mathbb{E}_{\mathbf{x}\sim\mathcal{D}_{\mathbf{x}}}\left[\int \pi_{\boldsymbol{\theta}}(\mathbf{y}_l|\mathbf{x})\nabla_{\boldsymbol{\theta}}\log\pi_{\boldsymbol{\theta}}(\mathbf{y}_l|\mathbf{x})d\mathbf{y}_l\right]$$

$$= \frac{1}{\beta}\mathbb{E}_{(\mathbf{x},\mathbf{y}_w,\mathbf{y}_l)\sim\bar{\mathcal{D}}_{\text{DPO}}}[\hat{r}_{\boldsymbol{\theta}}(\mathbf{x},\mathbf{y}_l)\nabla_{\boldsymbol{\theta}}\log\pi_{\boldsymbol{\theta}}(\mathbf{y}_l|\mathbf{x})],$$

where the last quality is using Eq. (20) again. We finish the proof. Note that the term $(\hat{r}_{\boldsymbol{\theta}}(\mathbf{x},\mathbf{y}_l) - \hat{r}_{\boldsymbol{\theta}}(\mathbf{x},\mathbf{y}_w))$ is likely to be positive since $\mathbf{y}_l$ is generated from $\pi_{\boldsymbol{\theta}}(\cdot|\mathbf{x})$.

## B Implementation Details

### B.1 Dataset Details

As explained in the section 5.1, we use a reward model to annotate all the completion samples. The quality score distributions of the used datasets in the paper are given in Table 5.

Some may doubt the reliance on reward model (RM) scores as quality criteria, given known calibration limitations that prevent these scores from being perfect human preference estimators. We justify this design through two principled arguments. First, our experiments require aggregating responses from heterogeneous sources, where a unified quantitative metric becomes indispensable for quality-aware sample reorganization. Second, empirical evidence from our LLM-as-a-judge comparison validates RMs practical superiority: on the Tulu3 UltraFeedback dataset, RM and LLM evaluators disagreed on response quality rankings for 10,000 prompts. Crucially, when training DPO models on datasets filtered by each method, RM-based selection achieved better performance, demonstrating its operational robustness. Recent work has also revealed the possible limitation of LLM-as-a-judge methods (Li et al., 2025). Although we do not claim RM scores are intrinsically perfect, their empirical stability and cross-domain applicability make them a functionally optimal choice for our multi-source quality stratification objectives.

### B.2 On-policy Data Generation

In the context of Direct Preference Optimization (DPO), on-policy data refers to preference pairs generated using the policy model that is currently being trained or a recent checkpoint thereof.

| Dataset | Type | Statistical Measures | | | | | | |
|---|---|---|---|---|---|---|---|---|
| | | **Mean** | **Std** | **Min** | **25%** | **Med** | **75%** | **Max** |
| OA2 Best+Worst | C | -3.00 | 3.02 | -20.9 | -4.80 | -3.30 | -1.66 | 19.4 |
| | R | -8.03 | 3.84 | -24.4 | -10.1 | -7.44 | -5.40 | 7.31 |
| OA2 Best+Low | C | -3.00 | 3.02 | -20.9 | -4.80 | -3.30 | -1.66 | 19.4 |
| | R | -7.01 | 2.91 | -21.6 | -8.94 | -6.97 | -4.91 | 6.19 |
| OA2 Best+Medium | C | -3.00 | 3.02 | -20.9 | -4.80 | -3.30 | -1.66 | 19.4 |
| | R | -5.32 | 3.06 | -22.0 | -6.91 | -5.12 | -3.48 | 13.9 |
| OA2 Best+High | C | -3.00 | 3.02 | -20.9 | -4.80 | -3.30 | -1.66 | 19.4 |
| | R | -4.80 | 2.77 | -22.0 | -6.36 | -4.72 | -3.14 | 13.9 |
| OA2 Low+Worst | C | -7.01 | 2.91 | -21.6 | -8.94 | -6.97 | -4.91 | 6.19 |
| | R | -8.03 | 3.84 | -24.4 | -10.1 | -7.44 | -5.40 | 7.31 |
| OA2 Medium+Worst | C | -5.32 | 3.06 | -22.0 | -6.91 | -5.12 | -3.48 | 13.9 |
| | R | -8.03 | 3.84 | -24.4 | -10.1 | -7.44 | -5.40 | 7.31 |
| OA2 High+Worst | C | -4.80 | 2.77 | -22.0 | -6.36 | -4.72 | -3.14 | 13.9 |
| | R | -8.03 | 3.84 | -24.4 | -10.1 | -7.44 | -5.40 | 7.31 |
| UF Best+Worst | C | -0.93 | 4.91 | -19.5 | 4.19 | -1.91 | 1.34 | 21.0 |
| | R | -8.54 | 3.36 | -24.5 | -10.8 | -8.56 | -6.19 | 6.28 |
| UF Best+Low | C | -0.93 | 4.91 | -19.5 | 4.19 | -1.91 | 1.34 | 21.0 |
| | R | -4.90 | 3.07 | -21.5 | -6.94 | -4.90 | -3.00 | 16.5 |
| UF Best+Medium | C | -0.93 | 4.91 | -19.5 | 4.19 | -1.91 | 1.34 | 21.0 |
| | R | -4.18 | 3.77 | -23.6 | -6.53 | -4.34 | -2.33 | 18.2 |
| UF Best+High | C | -0.93 | 4.91 | -19.5 | -4.19 | -1.91 | 1.34 | 21.0 |
| | R | -3.22 | 4.18 | -21.5 | -6.06 | -3.67 | -1.14 | 18.75 |
| UF Low+Worst | C | -4.90 | 3.07 | -21.5 | -6.94 | -4.90 | -3.00 | 16.5 |
| | R | -8.54 | 3.36 | -24.5 | -10.8 | -8.56 | -6.19 | 6.28 |
| UF Medium+Worst | C | -4.18 | 3.77 | -23.6 | -6.53 | -4.34 | -2.33 | 18.2 |
| | R | -8.54 | 3.36 | -24.5 | -10.8 | -8.56 | -6.19 | 6.28 |
| UF High+Worst | C | -3.22 | 4.18 | -21.5 | -6.06 | -3.67 | -1.14 | 18.75 |
| | R | -8.54 | 3.36 | -24.5 | -10.8 | -8.56 | -6.19 | 6.28 |
| UF LG-HQ | C | -2.98 | 3.70 | -21.5 | 5.28 | -3.34 | -1.32 | 18.2 |
| | R | -6.53 | 3.37 | -24.1 | -8.69 | -6.47 | -4.25 | 11.0 |
| UF LG-LQ | C | -5.16 | 3.67 | -23.6 | -7.50 | -5.34 | -3.22 | 18.2 |
| | R | -8.61 | 3.35 | -24.2 | -10.8 | -8.62 | -6.28 | 6.28 |
| UF SG-HQ | C | -3.56 | 3.76 | -23.6 | -5.84 | -3.80 | -1.78 | 18.2 |
| | R | -4.90 | 3.07 | -21.5 | -6.94 | -4.91 | -3.00 | 16.5 |
| UF SG-LQ | C | -6.59 | 3.32 | -23.6 | -8.81 | -6.56 | -4.34 | 14.9 |
| | R | -8.08 | 3.41 | -24.2 | -10.3 | -8.12 | -5.69 | 11.3 |
| UF HQ-On-Pol.10% | C | -0.93 | 4.91 | -19.5 | -4.19 | -1.91 | 1.34 | 21.0 |
| | R | -4.90 | 3.07 | -21.5 | -6.94 | -4.91 | -3.00 | 16.5 |
| UF LQ-On-Pol.20% | C | -4.18 | 3.77 | -23.6 | -6.53 | -4.34 | -2.33 | 18.2 |
| | R | -8.54 | 3.36 | -24.2 | -10.8 | -8.56 | -6.19 | 6.28 |

Table 5: The overall dataset quality score distribution (chosen (C) vs rejected (R)). "OA2" and "UF" stand for `Open Assistant 2` and `UltraFeedback`, respectively. The LG-HQ-inv. dataset utilizes the LG-HQ's chosen part and SG-HQ's rejected part, while the SG-HQ-inv. uses the SG-HQ's chosen part and SG-LQ's rejected part. The datasets in the exposure bias experiments share the same chosen responses within the same chosen quality level, respectively.

Training with such data allows the model to learn from its own evolving capabilities. Several methods leverage on-policy data, including fully online approaches like online DPO (Guo et al., 2024) and online iterative DPO (Dong et al., 2024a). While potentially effective, these online methods often require frequent interaction with a preference judge (e.g., a human annotator or a reward model) during the training loop, which can significantly increase computational and annotation costs.

An alternative strategy, which balances the benefits of on-policy data with practical constraints, involves generating a batch of on-policy data offline before commencing or resuming DPO training. This generated data can then be mixed with existing off-policy datasets (Lambert et al., 2024; Deng et al., 2025). In this paper, we adopt a similar offline generation approach, closely following the methodology described by Lambert et al. (2024). The specific process for generating our on-policy preference data is detailed in Algorithm 1.

---

**Algorithm 1** On-Policy Data Generation Process

---

1: **Input:** SFT model checkpoint $\pi_{\text{SFT}}$, Reward model $r_\phi$, Offline preference dataset $D_{\text{offline}} = \{(p_i, y_{w,i}, y_{l,i})\}_{i=1}^N$, On-policy data ratio $\rho$, Generations per prompt $k$.
2: **Output:** Mixed preference dataset $D_{\text{mixed}}$.
3: Initialize $D_{\text{on-policy}} = \emptyset$.
4: Sample a subset of prompts $P_{\text{on}} \subseteq \{p_i\}_{i=1}^N$ such that $|P_{\text{on}}| = \lfloor \rho \times N \rfloor$.
5: Let $I_{\text{on}} = \{i \mid p_i \in P_{\text{on}}\}$ be the indices of the selected prompts.
6: Let $D_{\text{remaining}} = \{(p_i, y_{w,i}, y_{l,i}) \mid i \notin I_{\text{on}}\}$.
7: **for** each index $i \in I_{\text{on}}$ **do**
8:     Let $p = p_i$, $y_w = y_{w,i}$, $y_l = y_{l,i}$.
9:     Generate $k$ candidate responses $\{y'_j\}_{j=1}^k$ using $p$: $y'_j \sim \pi_{\text{SFT}}(\cdot|p)$.     ▷ Using specified sampling parameters
10:     Score each generated response: $s'_j = r_\phi(p, y'_j)$ for $j = 1, \ldots, k$.
11:     Identify the best on-policy response: $y'_{\text{best}} = \arg\max_{y'_j}\{s'_j\}$. Let $s'_{\text{best}} = r_\phi(p, y'_{\text{best}})$.
12:     Retrieve the score of the original preferred response: $s_w = r_\phi(p, y_w)$.
13:     **if** $s'_{\text{best}} > s_w$ **then**
14:         Add new preference pair $(p, y'_{\text{best}}, y_w)$ to $D_{\text{on-policy}}$. ▷ Replace original chosen response
15:     **else**
16:         Add new preference pair $(p, y_w, y'_{\text{best}})$ to $D_{\text{on-policy}}$. ▷ Replace original rejected response
17:     **end if**
18: **end for**
19: Combine the datasets: $D_{\text{mixed}} = D_{\text{remaining}} \cup D_{\text{on-policy}}$.
20: **return** $D_{\text{mixed}}$.

---

The generation process utilized multinomial sampling with parameters specified in Table 6. The reward model $r_\phi$ used for scoring the generated responses in step 9 of Algorithm 1 is the same reward model used to create the initial offline preference dataset $D_{\text{offline}}$.

| Parameter | Value |
|---|---|
| Sampling Method | Multinomial Sampling |
| Sampling Temperature | 0.6 |
| Max Generation Length | 1024 tokens |
| Responses per Prompt ($k$) | 8 |

Table 6: Parameters for On-Policy Response Generation.

## B.3 Training Hyperparameters

For all DPO experiments, we adopt the standard DPO training pipeline using the Huggingface framework with the following hyperparameters:

- **Optimizer**: AdamW ($\beta_1 = 0.9$, $\beta_2 = 0.99$) with no weight decay
- **Learning Rate**: Linear warmup with ratio $= 0.1$ to a peak of $5 \times 10^{-7}$, followed by cosine decay

- **Batch Size**: A global size of 32 via gradient accumulation over 4 steps
- **Duration**: 2 epochs
- **DPO Beta**: 0.1
- **Sequence Length**: 2048
- **Precision**: bfloat16

For the continual SFT training mentioned in Table 4, we adjust its peak learning rate to $1 \times 10^{-6}$ and AdamW $\beta_2 = 0.95$, and keep other optimizer and batch size parameters the same as the DPO setting.

## B.4 Evaluation

We leverage the Tulu3 evaluation pipeline except for AlpacaEval-2 to exclude the possible evaluation data leakage. For the AlpacaEval-2 assessment, we adopt a generation config of beam-search multinomial sampling with `num_beams=3` and `temperature=1.0`. For the other benchmarks, we use the default configuration as described in Lambert et al. (2024). The score metrics used in our experiments are presented in Table 7.

| Benchmark | Core Metric | Setting / Details |
| --- | --- | --- |
| LC-AE2 | Length-Controlled Win-Rate | Alpaca-Eval 2.0 version. 0-shot. |
| MMLU | Accuracy | 5-shot setting. |
| TruthfulQA | MC2 | 6-shot setting. |
| IFEval | Instruction-Following Accuracy | 0-shot setting. |
| GSM8K | Exact-Match Accuracy | 8-shot setting. |

Table 7: Overview of Evaluation Benchmarks and Metrics. All the evaluations are run with only 1 attempt, i.e., under the pass@1 setting.

## B.5 Compute Resources

All experiments were conducted on a server with 128 CPU cores, 1024 GB memory, 96 TB SSD storage and 8 NVIDIA H20 GPUs. Under these conditions, each training step in the experiments takes approximately 10 seconds.

Running the full set of evaluation benchmarks (excluding Alpaca-Eval) on a single GPU requires approximately 6 hours, and Alpaca-Eval evaluation times vary between 10 and 30 minutes per model, due to network fluctuations and API request limits.

