# OpenReview forum: "What Matters in Data for DPO?"
_NeurIPS.cc/2025/Conference — NeurIPS 2025 poster_

### Official Review · Reviewer_RvM1 · 2025-07-01

**Clarity:** 3
**Significance:** 3
**Originality:** 3
**Rating:** 3
**Confidence:** 3

**Summary:**

This paper analyzes how the distribution of preference data influences DPO, from both theoretical and empirical perspectives. The authors find that the quality of the chosen responses plays a dominant role in optimizing the DPO objective, and that online DPO effectively reduces to SFT on the chosen responses.

**Questions:**

1. Lines 35–38 are duplicated.

2. Line 208 should use $\pi_l$ instead of $\pi_w$.

**Ethical Concerns:**

["NO or VERY MINOR ethics concerns only"]

**Final Justification:**

Issues Resolved:
1. Whether pi is small
2. The increase of the chosen-rejected margin
3. The collapse of online DPO
4. Numerical results

Issues Not Resolved:

* On the Usefulness and Implications of the Specific Online DPO Variant. The authors argue that fixing the chosen response while resampling only the rejected response provides a clean setting to disentangle effects. However, my concern is that this setup is too narrow: it only shows a special case rather than giving insights into general online DPO. Thus, the claimed “non-trivial insight” mostly reiterates that chosen responses matter, without extending understanding to broader DPO mechanisms.

* Online DPO vs. Offline DPO vs. SFT. The authors try to reconcile the seeming contradiction by carefully distinguishing datasets and training stages. But this sidesteps my original concern: if offline DPO consistently outperforms SFT, and their variant of online DPO reduces to SFT, then offline DPO would likely outperform their online DPO variant. Their clarification doesn’t resolve the tension, it just reframes the comparison.

* On the Intuitiveness of the Conclusions. The authors defend their conclusion as surprising, but from my perspective, the claim that “chosen response quality matters more than rejected response quality or contrastiveness” feels intuitive. What’s missing is a quantitative measure of how much more. Without that, the contribution seems qualitative and unsurprising.

In summary, I am inclined to give a negative score, but since two other reviewers expressed positive opinions and I have not fully reviewed the proof process, I will lower my confidence.

**Limitations:**

See weakness

**Quality:**

3

**Strengths And Weaknesses:**

**Strengths:**
This paper investigates how the selection of *chosen* and *rejected* data in DPO affects performance, which is a very important question.

**Weaknesses:**
1. The paper points out that *“The quality of the rejected responses may not always be critical when $\pi(y_w|x)$ is small.”* Could the authors analyze the actual magnitude of $\pi(y_w|x)$ observed in their experiments? Is it always very small? After one epoch of training, could $\pi(y_w|x)$ increase? If so, shouldn’t DPO be trained for more epochs, yet in practice people usually train for only one epoch.

2. How should we interpret the fact that the chosen–rejected margin in DPO increases as training progresses? If it is equivalent to SFT, then the model should converge to the SFT solution.

3. Under Theorem 4.5, DPO is essentially doing SFT, so training the model for a long time should not cause collapse, but in contrast, online DPO can easily overfit and produce long repetitive outputs.

4. Table 1 should include SFT results on the *chosen* responses to support the paper’s core claim: *“the quality of chosen responses plays a dominant role in optimizing the DPO objective and online DPO effectively reduces to SFT on the chosen responses.”*
In Table 2, the LG-HQ and SG-HQ results show that larger gap sizes are better; similarly, LG-LQ and SG-LQ also show that larger gaps are better. This contradicts the authors’ claim that *“DPO optimization primarily operates through quality anchoring rather than margin maximization.”*

---

> ### Author Rebuttal · Authors · 2025-07-30
>
> ### **1. Whether $\pi(y_w|x)$ is small**
>
> We apologize if we misunderstood your original comment and have done our best to respond based on our current understanding. We had difficulty locating the exact sentence “The quality of the rejected responses may not always be critical when $\pi(y_w|x)$ is small.” The closest related statement we could identify appears around line 190, where we discuss the roles of the distributions $\pi_w(y|x)$ and $\pi_l(y|x)$, rather than the specific probability $\pi(y_w|x)$ of generating a particular response $y_w$.
> Our intended point was the following: if two rejected response distributions, $\pi'_l(y|x)$ and $\pi''_l(y|x)$, differ only in regions where $\pi_w(y|x)$ is small, then the corresponding likelihood ratios $\pi_w(y|x)/\pi'_l(y|x)$ and $\pi_w(y|x)/\pi''_l(y|x)$ may still be similar. As a result, the model may exhibit similar behavior or performance under both distributions, despite the differences in rejected samples.
>
> That said, following your suggestion, we carefully reviewed all the experiments we conducted. In most of our DPO runs, the probability assigned to the chosen response either fluctuates or decreases over training, which is consistent with observations reported in recent studies (e.g., [1], [2]). Therefore, we do not necessarily expect $\pi(y_w|x)$ to increase during training.
>
> Furthermore, determining the optimal number of training epochs for DPO is nontrivial. As far as we are aware, the number of epochs varies widely across implementations. In our experiments, we find that training for two epochs generally yields better performance than just one. Additionally, widely-used methods such as classic online DPO and DPO with curriculum learning typically involve multiple epochs of training. We believe this variability is closely tied to both the characteristics of the data and the compatibility of the base model.
>
> ### **2. The increase of the chosen-rejected margin**
>
> We fully agree that the chosen–rejected margin tends to increase as DPO training progresses. However, we do not think that this observation contradicts the fact that DPO can converge to the SFT solution under certain conditions. Convergence to the SFT solution inherently involves increasing the logits (or likelihood) of the positive samples. This, in turn, implicitly reduces the relative likelihood of responses not used for instruction, thereby increasing the chosen–rejected margin as well. Furthermore, in our numerical experiments under the online DPO setting, we also observe that the probability of generating the chosen response increases over training, which closely resembles the behavior typically associated with SFT.
>
> ### **3. The collapse of online DPO**
>
> We fully agree with the reviewer that online DPO can be prone to collapse. We would like to respond from two perspectives:
>
> First, the form of online DPO we study in Theorem 4.5 is a special case in which the chosen responses are of very high quality (e.g., written by human experts) and remain fixed throughout training, while only the rejected responses are updated and generated by the model. In contrast, standard online DPO typically generates both the chosen and rejected responses from the model itself, making it more likely to overfit and collapse. In our case, since the chosen responses are externally sourced and of consistently high quality, the risk of collapse is mitigated. We acknowledge that this distinction makes our setting narrower in scope, and we agree that your concern is highly relevant to the standard formulation of online DPO.
>
> Second, we would like to point out that similar instability can also occur in SFT. There are numerous community reports indicating that SFT checkpoints sometimes perform worse than their pre-trained counterparts, even producing nonsensical or repetitive outputs. This issue has been observed in widely used models such as LLaMA and Gemma, including their most recent versions. These observations highlight that SFT, too, is vulnerable to overfitting and collapse under certain conditions.
>
> ### **4. Numerical results**
>
> We sincerely appreciate your great comments on the numerical results.  First, we present the results of SFT on the chosen response, and compare them with the online DPO we studied in Theorem 4.5.
>
> | **Dataset** | **Configuration** | **LC-AE2** | **MMLU** | **IFEval** | **TruthfulQA** | **GSM8K** |
> |:------------|:------------------|:-----------|:---------|:---------|:---------------|:----------|
> | N/A | SFT Baseline | 12.7 | 62.1 | 74.3 | 46.8 | 76.8 |
> | Open Assistant 2 | Continual SFT | 18.7 | 60.4 | 71.5 | 46.9 | 78.7 |
> | Open Assistant 2 | Online-DPO | 19.0 | 60.6 | 71.8 | 47.5 | 78.6 |
> | UltraFeedback | Continual SFT | 35.8 | 61.6 | 74.1 | 57.1 | 79.5 |
> | UltraFeedback | Online-DPO | 37.6 | 62.0 | 74.5 | 58.0 | 79.7 |
>
> The table demonstrates that our online DPO setup yields performance comparable to that of SFT, consistent with the prediction of Theorem 4.5. Furthermore, by comparing the SFT results with those in the original Table 1, we observe that incorporating rejected responses does provide some performance benefit, though the improvement is relatively modest. We hope these additional results help address your concerns.
>
> Also, we agree with your observation that, based on Table 2, comparing LG-HQ with SG-HQ (and similarly, LG-LQ with SG-LQ) may suggest that larger gap sizes lead to better performance. However, we would like to highlight that when comparing LG-HQ and SG-HQ, the difference is not solely in the gap size—the quality of the chosen responses in LG-HQ is also higher than that in SG-HQ.
>
> To clarify this point, we present the revised Table 2 below. In the table, “Avg.Chs” and “Avg.Diff” denote the average quality score of the chosen responses and the average quality gap between the chosen and rejected responses, respectively. Δ Score represents the performance difference between two rows on the LC-AE2 benchmark.
>
> - **Part 1** of the table illustrates the comparisons we consider most informative—those that hold the gap approximately constant while varying the quality of the chosen response.
>
> - **Part 2** corresponds to the comparisons you suggested, where both the gap and the chosen response quality differ.
>
> For example, the Avg.Chs values for LG-HQ and SG-HQ are –2.98 and –3.56, respectively, showing a notable difference in chosen response quality. This indicates that the observed performance differences may stem from both the chosen response quality and the gap size. As a result, we believe it is not safe to attribute the performance gains solely to the gap size based on LG-HQ vs. SG-HQ or LG-LQ vs. SG-LQ.
>
> To further disentangle these effects, we constructed two additional datasets:
>
> - **LG-HQ-inverse**, which retains the high chosen response quality of LG-HQ but reduces the chosen–rejected gap.
>
> - **SG-HQ-inverse**, which maintains the chosen response quality of SG-HQ while increasing the gap.
>
> As shown in Part 3 of the table, increasing the gap alone does not significantly affect DPO performance, thereby supporting our claim that the quality of the chosen response plays a more dominant role.
>
> We hope these additional numerical results help clarify our conclusions and address your concerns.
>
> | **Configuration** | **Avg.Chs** | **Avg.Diff** | **LC-AE2** | **Δ Score** | **MMLU** | **IFEval** | **GSM8K** |
> |:------------------|:------------|:-------------|:-----------|:------------|:---------|:---------|:----------|
> | **Part 1: (Gap size is held constant)** ||||||| |
> | LG-HQ (High Quality) | -2.98 | 3.54 | 33.0 | **+8.7** | 64.9 | 76.9 | 81.2 |
> | LG-LQ (Low Quality)  | -5.15 | 3.45 | 24.3 |           | 61.9 | 73.8 | 80.5 |
> | LG-HQ (High Quality) | -3.56 | 1.34 | 28.4 | **+7.1** | 64.0 | 74.2 | 80.8 |
> | SG-LQ (Low Quality)  | -6.59 | 1.49 | 21.3 |           | 62.6 | 72.3 | 78.0 |
> | **Part 2:** ||||||| |
> | LG-HQ (Large Gap)    | -2.98 | 3.54 | 33.0 | **+4.6** | 64.9 | 76.9 | 81.2 |
> | SG-HQ (Small Gap)    | -3.56 | 1.34 | 28.4 |           | 64.0 | 74.2 | 80.8 |
> | LG-LQ (Large Gap)    | -5.15 | 3.45 | 24.3 | **+3.0** | 61.9 | 73.8 | 80.5 |
> | SG-LQ (Small Gap)    | -6.59 | 1.49 | 21.3 |           | 62.6 | 72.3 | 78.0 |
> | **Part 3: Effect of Gap (Chosen quality is identical)** ||||||| |
> | LG-HQ (Large Gap)        | -2.98 | 3.54 | 33.0 | **+1.4** | 64.9 | 76.9 | 81.2 |
> | LG-HQ-inv (Small Gap)    | -2.98 | 1.92 | 31.6 |           | 64.7 | 76.5 | 81.3 |
> | SG-HQ-inv (Large Gap)    | -3.56 | 4.51 | 29.2 | **+0.8** | 64.2 | 75.1 | 80.5 |
> | SG-HQ (Small Gap)        | -3.56 | 1.34 | 28.4 |           | 64.0 | 74.2 | 80.8 |
>
> ### **Reference:**
> [1] Razin, N., Malladi, S., Bhaskar, A., Chen, D., Arora, S., & Hanin, B. (2025). Unintentional unalignment: Likelihood displacement in direct preference optimization. ICLR 2025
>
> [2] Ren, Y., & Sutherland, D. J. (2025). Learning dynamics of llm finetuning. ICLR2025

---

> > ### Comment · Reviewer_RvM1 · 2025-08-06
> >
> > Thank you very much for your response. The numerical results have made me even more confused:
> >
> > * The results for Continual SFT are 18.7 and 35.8, but in Table 1, the Best+Worst scores are 20.9 and 36.5. Doesn’t this clearly indicate that DPO outperforms SFT? Moreover, several papers \[1, 2] have shown that Online DPO significantly outperforms DPO, whereas your experimental conclusion suggests the opposite.
> > * Although the experiments in Part 2 aim to show that Avg.Chs is important while Avg.Diff is not, many papers emphasize that DPO is better than SFT \[3], so I still question whether the conclusions in your paper hold true on real-world datasets.
> >
> > [1] Table 2. *Direct Language Model Alignment from Online AI Feedback*
> >
> > [2] Section 6.2 "Main Results: RLHF Benefits from Online and/or Pseudo Labelling Data" in *Iterative Preference Learning from Human Feedback: Bridging Theory and Practice for RLHF under KL-Constraint*
> >
> > [3] Figure 1 in *Unpacking DPO and PPO: Disentangling Best Practices for Learning from Preference Feedback*

---

> > > ### Author Response · Authors · 2025-08-06
> > > **Thank you for your questions.**
> > >
> > > Thank you for your great comments. We are sorry for the confusion. We fully agree with your observations regarding the performance differences among SFT, DPO, and online DPO. However, we believe our conclusions are **not in contradiction** with existing literature or conventional wisdom. Instead, our work aims to **complement prior findings** by providing new insights into *why and how* these methods work—particularly by isolating the key factors that drive performance improvements in practice.
> > >
> > > **Online DPO V.S. Offline DPO**
> > >
> > > The key clarification here is that the form of online DPO we investigate in this paper is a **special form**, in which the **chosen response is fixed** (typically a high-quality, human-written or verified fact response), and only the **rejected response is resampled** from the current model. This is not the general online DPO setting studied in most prior works. Our goal in focusing on this narrower version is to **disentangle the contributions** of the different components involved in online DPO.
> > >
> > > In standard online DPO, such as in the works you cited, multiple responses are resampled, and human annotators are asked to identify the best one. As the model improves over training, the quality of the responses naturally improve. If the offline DPO dataset is of lower quality, it is expected that online DPO would outperform it because of the improved data quality. However, it remains unclear *which factor*—the improved chosen response, sharper rejected responses, or the growing preference gap—is most responsible for the gains.
> > >
> > > In contrast, by fixing the chosen response and only resampling the rejected one, we isolate the effect of the **rejected samples and the margin**, while holding the chosen response quality constant. Theorem 4.5 shows that in this setting, DPO behaves similarly to SFT with an additional regularization term. This, together with the numerical results, suggests that the **primary driver of performance** is the quality of the chosen response, while the rejected response plays a more **secondary role**. This offers a possible explanation for why classical online DPO improves as training progresses: the improvement is largely due to the increasing quality of the chosen responses, rather than just widening the preference gap or refining the rejected responses. This is also the main message of our paper in the pure offline case.
> > >
> > > To be clear, we are **not claiming that classical online DPO underperforms offline DPO**. Rather, we are highlighting why online DPO performs well, and which aspects of the data are most responsible for that success.
> > >
> > > **DPO V.S. SFT**
> > >
> > > We completely agree with the well-established result that DPO outperforms SFT in many settings. Our conclusions do not contradict this. The core takeaway of our work is that **within DPO training, the quality of the chosen response matters more** than the gap or the rejected response quality. While our analysis of the special online DPO case reveals some similarities to SFT (due to the fixed, high-quality chosen response), we are not claiming that DPO in general is equivalent to or worse than SFT.
> > >
> > > Sorry that we may have not fully understood your confusion on DPO and SFT since in Part 2 we did not say anything about SFT. We would love to discuss more if our response does not resolve your concern or you want to clarify a bit. Thank you for your time in reviewing our paper.

---

> > > > ### Comment · Reviewer_RvM1 · 2025-08-08
> > > >
> > > > Thank you for your detailed rebuttal and for clarifying the specific setting of your online DPO experiments. I appreciate you clearing up my initial misunderstanding.
> > > >
> > > > I now understand that your analysis focuses on a specific form of online DPO where the chosen response is fixed and only the rejected response is resampled. While this clarifies the scope, it also leads me to believe that the paper's contribution may be narrower than I had initially perceived. If the primary claim is that this particular variant of online DPO behaves similarly to SFT, the impact of the finding seems more limited.
> > > >
> > > > Based on this new understanding, a point of potential tension arises in your argument, and I would appreciate your thoughts on it. You note that in your specific setting, "Theorem 4.5 shows that... DPO behaves similarly to SFT with an additional regularization term," which suggests the quality of the chosen response sets the performance ceiling. At the same time, you state, "We completely agree with the well-established result that DPO outperforms SFT in many settings."
> > > >
> > > > Juxtaposing these two points leads to a somewhat counter-intuitive conclusion: if your specific online DPO variant behaves like SFT, and general offline DPO is superior to SFT, it would imply that standard offline DPO might outperform the specific online DPO you study. This seems unusual, as one would expect an online method to hold some advantage.
> > > >
> > > > Furthermore, regarding your central message that "within DPO training, the quality of the chosen response matters more than the gap or the rejected response quality," I find this conclusion to be quite intuitive. The critical question, then, is not whether the chosen response is more important, but by how much. A qualitative claim that one factor is more important than another is not entirely surprising; the significance of this finding would be substantially greater if the paper provided a quantitative analysis of this difference. My initial score of 4 was based on the misunderstanding that your paper had demonstrated a surprising phenomenon in the general case.

---

> > > > > ### Author Response · Authors · 2025-08-08
> > > > > **Thank you for the further clarification**
> > > > >
> > > > > **1. On the Usefulness and Implications of the Specific Online DPO Variant**
> > > > >
> > > > > We fully understand your concern. While it is true that our analysis focuses on a specific variant of online DPO, we believe this variant is useful and offers valuable and non-trivial insights into the underlying mechanisms of online DPO. You also make the claim that  “one would expect an online method to hold some advantage”. But do we truly understand which specific components of online DPO contribute to the advantage? Is it the improved quality of both the chosen and rejected responses? Or is one more important than the other? Could it be the increased diversity of the sampled responses?
> > > > >
> > > > > Our **controlled variant** of online DPO, though narrower in scope, allows us to **disentangle some of these effects** more clearly and rigorously.
> > > > >
> > > > > One of the key findings from our analysis is that **contrastiveness between chosen and rejected responses may not be as essential** as often assumed. Contrastiveness has traditionally been viewed as a central ingredient behind the success of DPO and RLHF. However, in our setup, where the chosen response remains fixed and rejected responses gradually improve, the model's behavior increasingly resembles SFT. This observation highlights that the quality of the chosen response, rather than the contrastiveness per se, may be the dominant factor driving performance. We view this as a **non-trivial insight** that challenges common assumptions and contributes to a deeper understanding of DPO’s mechanisms.
> > > > >
> > > > > In summary, while our variant may not cover all aspects of general online DPO, we believe it serves as a **valuable surrogate** that enables a more **principled and interpretable analysis**. The insights derived from this controlled setting help shed light on the success of online DPO in practice, and we hope they can inspire further theoretical and empirical work on the broader formulation.
> > > > >
> > > > > **2. Online DPO V.S. Offline DPO V.S. SFT**
> > > > >
> > > > > We now fully understand the source of your concern. Every statement you made is correct, but we believe the apparent tension arises from different assumptions about the data and training stages involved. Below, we break this down more carefully.
> > > > >
> > > > > - “DPO outperforms SFT.” We agree with this widely observed result—but we want to clarify what we mean by it. Our statement refers to the common practice of applying DPO after an initial SFT stage. This combined approach (SFT + DPO) consistently improves performance relative to SFT alone.
> > > > > Importantly, SFT and DPO are usually trained on different datasets. SFT typically uses supervised data (e.g., human-written completions), while DPO relies on preference-labeled data. However, if the reviewer was referring to comparing DPO and SFT when applied independently, i.e., training with DPO from scratch without prior SFT, we agree this is a different question—and one for which there may not yet be a definitive consensus. Therefore, we interpret “DPO outperforms SFT” as shorthand for:
> > > > >
> > > > > $\qquad\qquad$ **SFT (on dataset A)+DPO (on preference dataset B)>SFT (on dataset A)**.
> > > > >
> > > > >
> > > > > - “Our specific online DPO behaves like SFT.”
> > > > >  Under our online DPO setup, the training dynamics of online DPO resemble those of SFT applied to the same fixed chosen responses—with an additional regularization term.
> > > > > To be clear, we are not claiming that our specific online DPO is equivalent to the huge SFT before the DPO/RLHF step (which often involves larger-scale, more diverse data). Instead, our comparison is with:
> > > > >
> > > > > $\qquad\qquad$ **Online DPO (fixed chosen responses)≈SFT (on the same fixed chosen responses).**
> > > > >
> > > > > Reconciling the Apparent Contradiction. The concern seems to be that: **Our Online DPO≈SFT<Offline DPO** might suggest that offline DPO outperforms an online method, which seems counterintuitive. But if we expand this into the full training context, the relationship becomes clearer:
> > > > >
> > > > > **Online DPO (fixed chosen responses) ≈ SFT (on the same fixed chosen responses) < SFT (on the same fixed chosen responses)+Offline DPO (new preference data).**
> > > > >
> > > > > In other words, if we want to apply “DPO outperforms SFT” as we discussed, offline DPO is applied on new, additional data, while our online variant is effectively SFT-style training on fixed responses.
> > > > >
> > > > > - "One would expect an online method to hold some advantage." We also agree with this statement in principle. However, this is actually a tricky statement. For instance, consider a 7B model doing online DPO VS a 7B model fine-tuned using much more offline DPO data generated by a much stronger 180B model. Without careful control of data and evaluation criteria, comparing online and offline DPO may not be informative. That’s why our specific online DPO setup is useful: by fixing the chosen responses, we remove one major variable and focus only on the rejected responses. We can then construct matched offline datasets using the same chosen responses, making the comparison fair and interpretable.

---

> > > > > > ### Author Response · Authors · 2025-08-08
> > > > > > **Continued Response**
> > > > > >
> > > > > > **On the Intuitiveness of the Conclusions**
> > > > > >
> > > > > > On the one hand, we are encouraged to hear that you find our main conclusion consistent with your intuition. On the other hand, from our own perspective, this result was **not obvious before we began this work.**
> > > > > >
> > > > > > To illustrate: in the RLHF pipeline, the first step is to train a reward model from preference labels. When the labels are accurate, we expect to obtain a “good” reward model, and the prevailing view is that its quality depends primarily on the  the **correctness of the preference labels** and **contrastiveness of the two responses**. The absolute quality of the chosen response is rarely emphasized as the key driver. Given the intrinsic connections between RLHF and DPO, it was reasonable to approach DPO with the same assumption.
> > > > > >
> > > > > > Similarly, regarding your comment that “one would expect an online method to hold some advantage,” again,  the question is where that advantage comes from. Before this work, we would have expected contrastiveness to play a significant role. Our results show that, contrary to that intuition, **contrastiveness plays a much smaller role** in general DPO settings, regardless of online or offline, than previously assumed. From our perspective, this was a surprising and non-trivial finding. Another common view is that the online DPO can get better and better data as the process goes. But it is unknown that whether the improvement of the chosen and the rejected responses has the same role. Our results indicate that the online DPO's success stems mainly from the improvement of the chosen responses. The rejected responses only play a secondary role. **The non-effectiveness of the quality of the rejected response** is also surprising to us.
> > > > > >
> > > > > > As for your question of by how much the chosen response matters more, we have presented quantitative results across multiple tasks.  In our experiments the absolute improvements differ across the five tasks. The magnitude of the effect varies with the task, dataset, and model. It does not seem meaningful to have an aggregated quantitative results on all tasks, all models and all datasets. However, the qualitative pattern—that chosen response quality dominates over gap size or rejected response quality—holds consistently across all five tasks, 14 configurations, and two datasets.  We view the robustness of this qualitative conclusion across varied settings as a meaningful and challenging result.
> > > > > >
> > > > > > We hope our clarification here can help! We really appreciate your time and patience. Thank you for these great comments!

---

### Official Review · Reviewer_fHiL · 2025-07-02

**Clarity:** 2
**Significance:** 4
**Originality:** 4
**Rating:** 5
**Confidence:** 4

**Summary:**

This paper provides a theoretical analysis of how the distribution of preference data influences DPO training. The key insight is that the quality of the chosen response plays a central role, whereas the quality of the rejected response and the degree of contrastiveness between the two responses appear to be less critical. The empirical results presented in the paper support these theoretical findings.

**Questions:**

See Strengths and Weaknesses.

**Ethical Concerns:**

["NO or VERY MINOR ethics concerns only"]

**Final Justification:**

I had two major concerns about the paper. First, the empirical results lacked clarity, say, the metrics being reported were not clearly specified. Second, the example used to support the claim that "the quality of the rejected responses may not always be critical" was not appropriate. Both concerns have been addressed by the authors.

I believe the theoretical contributions of the paper offer novel insights into DPO which could provide practical guidance on how to design effective training data for it. Since my main concerns have been resolved, I decide to increase my score.

**Limitations:**

See Strengths and Weaknesses.

**Quality:**

3

**Strengths And Weaknesses:**

**Major comments:**

The paper is theoretically solid and well-supported by empirical results. It offers a novel and insightful perspective on how preference data influences DPO training.

The theoretical part is well-written and well-structured. One minor suggestion is that it would be helpful to include a concrete example of of $\pi_w$ and $\pi_l$, say $\pi_w(y\mid x) = \pi_{\text{ref}}(y\mid x) \exp\left(r(x, y)/\beta\right)$ and $\pi_l = \pi_{\text{ref}} \exp\left(-r(x, y)/\beta\right)$, to help readers better understand how these distributions reflect response quality and help avoid potential confusion between $\pi_w(y), \pi_l(y)$ and $\pi_{\text{ref}}(y_w), \pi_{\text{ref}}(y_l)$. Besides, I’m not fully convinced by the example mentioned in line 198-204, which is used to support the claim that "the quality of rejected responses may not always be critical." Since the likelihood ratio $\pi_w(y)/\pi_l(y)$ between best vs worst and best vs random is different, especially when the reward is extremely high.

Regarding the empirical section, additional clarity is needed. It is not explicitly stated what the numbers in the tables represent. I assume they reflect expected rewards, but this should be made explicit. For GSM8K, metrics such as pass\@1, pass\@k, or major\@k are more standard.

**Minor comments:**

* Line 208: One of the occurrences of $\pi_w$ should likely be $\pi_l$.
* Line 216: The word "importantconsistent" should be split into two words.

---

> ### Author Rebuttal · Authors · 2025-07-30
>
> ### **1. More details on the empirical section**
>
> We sincerely thank the reviewer for the great comments and suggestions.
> We will make sure that in the revised version, we will include the exact metric of each benchmark used in our experiments. And the following is for your reference.
>
> | Benchmark | Core Metric | Setting / Details |
> | :--- | :--- | :--- |
> | LC-AE2 | Length-Controlled Win-Rate | Alpaca-Eval 2.0 version. 0-shot. |
> | MMLU | Accuracy | 5-shot setting. |
> | TruthfulQA | MC2 | 6-shot setting. |
> | IFEval | Instruction-Following Accuracy | 0-shot setting. |
> | GSM8K | Exact-Match Accuracy | 8-shot setting. |
>
> *Caption: Overview of Evaluation Benchmarks and Metrics. All the evaluations are run with only 1 attempt, i.e., under the pass@1 setting.*
>
> ### **2. Concrete examples of $\pi_w$ and $\pi_l$**
> We greatly appreciate your valuable comments and will follow your excellent example in the revised version. We also recently found similar ideas of your example in prior work [1], which we will cite and discuss in the updated manuscript.
>
> ### **3. Example in line 198-204**
> We sincerely apologize for any confusion our presentation may have caused. Our intention was to use the example in lines 198–204 as **numerical** support, rather than very rigor theoretical justification, for the claim that "the quality of rejected responses may not always be critical." We fully agree that the likelihood ratios $\pi_w(y)/\pi_l(y)$ between best vs. worst and best vs. random can indeed differ. However, despite our efforts, deriving analytical results to characterize these differences seems to be challenging. We will highlight this as an important direction for future work.
>
>
> **Reference:**
>
> [1] Feng, Y., Kwiatkowski, A., Zheng, K., Kempe, J., & Duan, Y. (2025). Pilaf: Optimal human preference sampling for reward modeling. arXiv preprint arXiv:2502.04270.

---

> > ### Comment · Reviewer_fHiL · 2025-08-02
> >
> > For your reference, the underlying distribution of best-of-n sampling has been established by some existing works [1,2]. Worst-of-n’s distribution can be derived analogously. The difference between likelihood ratios of best/worst and best/random can become extremely large, especially in the high-reward regime. I just feel like this example does not support your claim and you need to use it with caution.
> >
> > [1] Beirami, A., Agarwal, A., Berant, J., D'Amour, A., Eisenstein, J., Nagpal, C. and Suresh, A.T., 2024. Theoretical guarantees on the best-of-n alignment policy. arXiv preprint arXiv:2401.01879.
> >
> > [2] Gui, L., Gârbacea, C. and Veitch, V., 2024. Bonbon alignment for large language models and the sweetness of best-of-n sampling. Advances in Neural Information Processing Systems, 37, pp.2851-2885.

---

> > > ### Author Response · Authors · 2025-08-04
> > > **Thank you for the further comment!**
> > >
> > > We sincerely thank you for pointing us to these two important papers. We apologize for not being more familiar with this line of theoretical work at the time of writing. After carefully reviewing the references during the past two days, we now fully understand your concern. To avoid any potential misinterpretation, we will remove the example in question from the revised version of the paper. Given that our core claims are already supported by extensive numerical evidence, we believe this adjustment will strengthen the overall clarity and rigor of our presentation. Thank you again for your thoughtful and constructive feedback.

---

> > > > ### Comment · Reviewer_fHiL · 2025-08-04
> > > >
> > > > Thank you for the response. I think most of my concerns have been addressed. I will increase my score.

---

### Official Review · Reviewer_jmf2 · 2025-07-03

**Clarity:** 1
**Significance:** 3
**Originality:** 3
**Rating:** 5
**Confidence:** 3

**Summary:**

The authors talk about what matters in data for DPO with some theories and experimentations. From the theory side, the authors show that where the  high-quality chosen responses dominate the gradient, and when the samples are generated on policy, DPO becomes SFT. The experimentations try to support the claims such as the chosen sample quality is the only thing that matters in DPO, and preference gap and exposure bias, which are commonly believed to be important given past literature, also have limited impact.

**Questions:**

see weakness

**Ethical Concerns:**

["NO or VERY MINOR ethics concerns only"]

**Final Justification:**

The new experiments, analysis, and practical guide addressed most of my concerns. I am willing to give an accept for this paper.

**Limitations:**

yes

**Quality:**

2

**Strengths And Weaknesses:**

Strength:
1. This is an interesting and important topic to have both theoretical insights and experiments to support it. The paper offers both and provides good insights for practitioners to construct DPO data when training their models.

Weakness:
1. The biggest concern and weakness is that, although the axis of experiments design was promising, specifically “Fixed Chosen, Varied Rejected” and “Fixed Rejected, Varied Chosen.”, the experiment results are not comprehensive enough to support the theories. I would like to see a more informative analysis that includes additional quality levels, such as at least 3 different levels of responses for chosen and rejected (9 experiments in total). Currently table 1 only has 3 rows, which is insufficient to reach the conclusions the author claimed.
2. Regarding theorem 4.5, I would like to see experimentation results to support it, which shows that the performance of the SFT results vs the DPO with on-policy samples.
3. Regarding Table 2, it does seem that there is some difference between LG-HQ-inv & LG-HQ, SG-HQ & SG-HQ-inv. I think the main problem is still the lack of comprehensive experiments to draw strong conclusions about the role of gap size versus chosen quality, though I think this table is already better than Table 1.
4. The experiments' terminologies are confusing and introduced unclearly. For example, for Table 1, the caption is telling me “The “Type” column abbreviates combinations of chosen and rejected sample quality, where “Best” and “Worst” denote the quality of chosen and rejected samples, respectively, and “High” or “Low” indicates the quality of the paired response.” which is unclear what those levels refer to. It would be great if the terminologies are specified in a clearer way.

---

> ### Author Rebuttal · Authors · 2025-07-30
>
> ### **1. More comprehensive numerical experiments**
> Thank you very much for your constructive suggestion to conduct more comprehensive numerical experiments. Following your recommendation, we paired each prompt in both the OpenAssistant 2 and UltraFeedback datasets with five responses of varying quality, labeled as *best, high, medium, low*, and *worst*. For each dataset, we conducted two sets of experiments: (1) fixing the best response as the chosen one and varying the rejected response among the remaining four, and (2) fixing the worst response as the rejected one and varying the chosen response.
>
> #### **Open Assistant 2: fix the best as the chosen and vary the quality of the rejected.**
>
> | **Dataset** | **Configuration** | **GSM8K** | **LC-AE2** | **MMLU** | **IFEval** | **TruthfulQA** |
> |:------------|:------------------|:----------|:----------|:---------|:---------|:---------------|
> | Open Assistant 2 (Fixed Best) | Best/Worst | 78.4      | 20.9      | 62.8     | 72.3     | 48.4           |
> | Open Assistant 2 (Fixed Best) | Best/Low  | 78.6      | 19.2      | 62.6     | 72.7     | 47.4           |
> | Open Assistant 2 (Fixed Best) | Best/Medium | 79.3      | 19.3      | 62.8     | 71.4     | 49.1           |
> | Open Assistant 2 (Fixed Best) | Best/High | 78.4      | 19.6      | 62.7     | 72.1     | 47.5           |
>
> The performance is actually quite similar across all five tasks, suggesting that the quality of the rejected response may not be a critical factor. This also implies that the chosen–rejected quality gap may not be fundamentally important. For instance, on GSM8K and TruthfulQA, using the medium-quality response as the rejected one yields the best performance. Similarly, on IFEval, using low-quality responses as the rejected ones leads to slightly better results.
>
> #### **Open Assistant 2: fix the worst as the rejected and vary the quality of the chosen.**
>
> | **Dataset** | **Configuration** | **GSM8K** | **LC-AE2** | **MMLU** | **IFEval** | **TruthfulQA** |
> |:------------|:------------------|:----------|:----------|:---------|:---------|:---------------|
> | Open Assistant 2 (Fixed Worst) | Low/Worst | 77.5      | 15.2      | 61.2     | 66.5     | 47.1           |
> | Open Assistant 2 (Fixed Worst) | Medium/Worst | 78.2      | 17.0      | 61.3     | 70.4     | 48.0           |
> | Open Assistant 2 (Fixed Worst) | High/Worst | 78.2      | 17.4      | 61.2     | 70.2     | 47.6           |
> | Open Assistant 2 (Fixed Worst) | Best/Worst | 78.4      | 20.9      | 62.8     | 72.3     | 48.4           |
>
> When the chosen response is fixed to be of the best quality, the model consistently achieves the highest performance across all tasks. This highlights the critical role that the quality of the chosen response plays in driving learning.
>
> #### **UltraFeedback: fix the best as the chosen and vary the quality of the rejected.**
>
> | **Dataset** | **Configuration** | **GSM8K** | **LC-AE2** | **MMLU** | **IFEval** | **TruthfulQA** |
> |:------------|:------------------|:----------|:----------|:---------|:---------|:---------------|
> | UltraFeedback (Fixed Best) | Best/Worst | 80.4      | 36.5      | 64.8     | 77.4     | 62.2           |
> | UltraFeedback (Fixed Best) | Best/Low  | 80.8      | 34.5      | 63.4     | 76.9     | 58.6           |
> | UltraFeedback (Fixed Best) | Best/Medium | 80.2      | 34.2      | 63.3     | 76.7     | 59.4           |
> | UltraFeedback (Fixed Best) | Best/High | 79.0      | 33.6      | 62.5     | 76.0     | 58.7          |
>
> Similarly, the performance does not exhibit a monotonic trend as the quality of the rejected response increases or decreases. This confirms that the quality of the rejected sample alone may not be a reliable indicator of DPO performance.
>
> #### **UltraFeedback: fix the worst as the rejected response and vary the quality of the chosen response.**
> | **Dataset** | **Configuration** | **GSM8K** | **LC-AE2** | **MMLU** | **IFEval** | **TruthfulQA** |
> |:------------|:------------------|:----------|:----------|:---------|:---------|:---------------|
> | UltraFeedback (Fixed Worst) | Low/Worst | 79.3      | 25.8      | 61.4     | 76.5     | 56.1           |
> | UltraFeedback (Fixed Worst) | Medium/Worst | 78.6      | 26.7      | 62.1     | 75.8     | 58.0           |
> | UltraFeedback (Fixed Worst) | High/Worst | 79.5      | 30.9      | 63.7     | 77.0     | 61.3           |
> | UltraFeedback (Fixed Worst) | Best/Worst | 80.4      | 36.5      | 64.8     | 77.4     | 62.2           |
>
> The set of experiments show that increasing the quality of the chosen response leads to a consistent, monotonic improvement in DPO training performance.
>
> However, when we fix the worst-quality response as the rejected one, increasing the quality of the chosen response not only improves the quality of the chosen sample itself but also enlarges the gap between the chosen and rejected responses. To further disentangle the effect of this quality gap from the absolute quality of the responses, we design an additional set of experiments. Specifically, we construct four new datasets—**LG-HQ, LG-LQ, SG-HQ, and SG-LQ**. The LG-HQ and LG-LQ datasets have similarly large chosen–rejected gaps but differ in the absolute quality of their responses. In contrast, SG-HQ and SG-LQ have similarly small gaps but also differ in response quality.
>
> | **Configuration** | **Avg.Chs** | **Avg.Diff** | **LC-AE2** | **Δ Score** | **MMLU** | **IFEval** | **GSM8K** |
> |:------------------|:------------|:-------------|:-----------|:------------|:---------|:---------|:----------|
> | LG-HQ (High Quality) | -2.98 | 3.54 | 33.0 | **+8.7** | 64.9 | 76.9 | 81.2 |
> | LG-LQ (Low Quality)  | -5.15 | 3.45 | 24.3 | | 61.9 | 73.8 | 80.5 |
> | SG-HQ (High Quality) | -3.56 | 1.34 | 28.4 | **+7.1** | 64.0 | 74.2 | 80.8 |
> | SG-LQ (Low Quality)  | -6.59 | 1.49 | 21.3 | | 62.6 | 72.3 | 78.0 |
>
> “Avg.Chs” and “Avg.Diff” refer to the average quality score of the chosen responses and the average quality gap between the chosen and rejected responses, respectively. The Δ Score is the performance difference between the HQ and the LQ on LC-AE2. Across all four tasks, we observe consistently better performance when the chosen responses are of higher quality. Moreover, the magnitude of improvement is comparable across both large-gap and small-gap settings (8.7 vs. 7.1, 3.0 vs. 1.4, 3.1 vs. 1.9, 0.7 vs. 1.2), demonstrating the generalizability of this observation regardless of the gap size.
>
> To double check our findings on the non-essential effect of the quality gap, we construct two additional datasets. The first, **LG-HQ-inverse**, retains the high chosen response quality of LG-HQ but reduces the chosen–rejected gap. The second, **SG-HQ-inverse**, preserves the high chosen response quality of SG-HQ while increasing the gap.
>
> | **Configuration** | **Avg.Chs** | **Avg.Diff** | **LC-AE2** | **Δ Score** | **MMLU** | **IFEval** | **GSM8K** |
> |:------------------|:------------|:-------------|:-----------|:------------|:---------|:---------|:----------|
> | LG-HQ (Large Gap)       | -2.98 | 3.54 | 33.0 | **+1.4** | 64.9 | 76.9 | 81.2 |
> | LG-HQ-inv (Small Gap) | -2.98 | 1.92 | 31.6 | | 64.7 | 76.5 | 81.3 |
> | SG-HQ-inv (Large Gap) | -3.56 | 4.51 | 29.2 | **+0.8** | 64.2 | 75.1 | 80.5 |
> | SG-HQ (Small Gap)       | -3.56 | 1.34 | 28.4 | | 64.0 | 74.2 | 80.8 |
>
> Expanding the gap does lead to some performance improvement, but the gains are notably limited compared to those achieved by improving the quality of the responses. In fact, on GSM8K, we even observe a slight decrease in performance when the gap is increased.
>
> We hope these additional results address your concerns.
>
> ### **2. The numerical performance of SFT and online DPO**
>
> Here, we present the numerical results supporting Theorem 4.5, including evaluations on two datasets. We find that continual SFT and online DPO (with a fixed chosen response) exhibit similar performance across all five tasks. We sincerely thank you for this insightful comment, which has helped us improve the comprehensiveness of our experimental evaluation.
>
> | **Dataset** | **Configuration** | **LC-AE2** | **MMLU** | **IFEval** | **TruthfulQA** | **GSM8K** |
> |:------------|:------------------|:-----------|:---------|:---------|:---------------|:----------|
> | N/A | SFT Baseline | 12.7 | 62.1 | 74.3 | 46.8 | 76.8 |
> | Open Assistant 2 | Continual SFT | 18.7 | 60.4 | 71.5 | 46.9 | 78.7 |
> | Open Assistant 2 | Online-DPO | 19.0 | 60.6 | 71.8 | 47.5 | 78.6 |
> | UltraFeedback | Continual SFT | 35.8 | 61.6 | 74.1 | 57.1 | 79.5 |
> | UltraFeedback | Online-DPO | 37.6 | 62.0 | 74.5 | 58.0 | 79.7 |
>
> ### **3. Terminologies**
> Thank you for your great catches. We will carefully update them in our next version.

---

> ### Comment · Reviewer_jmf2 · 2025-08-04
>
> Thanks for the additional experiments!
>
> Based on the UltraFeedback fix the best experiment, it seems that as the quality of reject response decreases, the performance monotonically increases besides just 2 exceptions? (Medium/Worst GSM8K & IFEval). And for fix worst, I am not surprised of the results as a new finding because the preferred response intuitively serves as the upper bound of the performance.
>
> For the SFT & DPO comparison experiment, it seems that DPO appears to perform noticeably better than continual SFT. What’s your take on that? So this does seem that DPO is doing something better than SFT?
>
> One final question is, since the paper is centered on “what matters in data for DPO,” what message does it convey for practitioners constructing DPO datasets?  In practice, we samples responses from the model without knowing their absolute qualities, and we query human to annotated the preference dataset. In some sense there is no control of the qualities generated by the model. For example, we cannot force the model to generate high quality response, and if the model is able to generate high quality response, there is no reason for us to not use the high quality response but rather use some low quality response as the chosen one. Plus, it is hard to evaluate about the absolute scores, which is why people use preference learning methods such as DPO. If absolute scores are given, you should use RL then. Thus, the practical limitations would be my biggest concern about the impact of the paper.

---

> > ### Author Response · Authors · 2025-08-04
> > **Thank you for the great comments!**
> >
> > We sincerely thank the reviewer for the careful reading and thoughtful follow-up comments. Please find our detailed response below.
> >
> > ### **1. On the role of the preference gap**
> > We appreciate your observations that, in the UltraFeedback with the fixed best setting, performance appears to increase monotonically as the quality of the rejected decreases in many cases. We also acknowledge that the preference gap can contribute to DPO performance. However, our main claim is that when compared to the influence of the chosen response quality, the effect of the gap is relatively mild. We would like to highlight a few supporting numerical results:
> > - In the last two tables under the first point of our previous response, we show that widening the gap can yield a modest improvement. For example, comparing LG-HQ vs. LG-HQ-inv and SG-HQ-inv vs. SG-HQ results in a performance gain of roughly **1 point** on LC-AE2. In contrast, when we hold the gap constant and increase the chosen response quality, the performance can improve by over **7 points** on LC-AE2. This trend is consistently observed across all four tasks in the table.
> > - Moreover, on GSM8K, increasing the preference gap actually leads to a slight performance drop. In OpenAssistant2 with the fixed best setting (first table), the relationship between preference gap and performance is even less clear and appears non-monotonic.
> >
> > In summary, while we agree that the preference gap can help in many cases, we argue that it seems to have a secondary effect and is not a consistently reliable indicator of performance.
> >
> > ### **2. On the better performance of online DPO than continual SFT**
> >
> > We agree with your observation that online DPO performs slightly better than continual SFT in our experiments. However, we would like to emphasize that, from our perspective, the performance difference is **relatively modest**. In most cases, the gap is around 0.5 points, with the largest difference being 1.8 points on LC-AE2 with UltraFeedback.
> >
> > Our interpretation is as follows: while the chosen response serves as the primary learning signal, similar to SFT, the rejected response offers a secondary but valuable signal. It introduces a mild negative gradient that helps the model move away from less desirable outputs, something SFT alone does not provide. Moreover, as shown in Theorem 4.5, online DPO includes an **extra regularization term** compared to SFT. This term may contribute to **better generalization** and help **prevent overfitting**, potentially explaining some of the performance gains we observe.
> >
> > ### **3. On the practical implications for data management**
> >
> > Thank you very much for your insightful and practically relevant comment. Actually, the managerial insights are exactly what we want to emphasize. It is true that practitioners often sample responses from a model and rely on human annotators to provide relative preferences. However, our work still offers several concrete and actionable takeaways for improving the effectiveness of DPO training:
> >
> > - **Data Rewriting Strategy**: As you correctly point out, we cannot force a model to generate high-quality responses. In response to this challenge, a growing trend in industry practice is to allow annotators to **rewrite** or **refine** responses. For example, **Dataloop**, a company specializing in RLHF data services, enables response rewriting in its RLHF Studio.
> > Of course, involving human annotators in rewriting responses is **costly**. This raises an important managerial question: under a fixed annotation budget, how should resources be allocated most effectively? Our findings suggest that annotation efforts should be prioritized toward improving the quality of chosen responses, particularly when the initial chosen responses are far below human-level quality. In contrast, investing effort in modifying rejected responses or merely increasing the preference gap could be less helpful.
> >
> > - **Data Curation Strategy**:  Currently, preference pairs with small margins are often discarded. However, our results indicate that if the **chosen response is of high absolute quality**, then such pairs still provide useful training signal. Thus, we recommend a shift in mindset: instead of asking “How much better is the winner?”, data curators should ask “Is the winner a high-quality response?” to improve the data efficiency.
> >
> > - **Guidance for Response Sampling**: Best-of-$k$ sampling is a widely used approach trying to control the qualities of the generated responses, and it's an important task to choose the best $k$. Our analysis suggests that the primary benefit of increasing $k$ is in improving the quality of the chosen response, not in producing a more extreme or diverse rejected response. This insight implies that once increasing $k$ no longer significantly improves the best response in the sample set, there is little value in continuing to increase $k$.
> >
> > We hope that our explanation here could resolve your concern.

---

> > > ### Comment · Reviewer_jmf2 · 2025-08-04
> > >
> > > Thanks for the detailed response! I think this addressed most of the concerns I have. I have raised my score to an accept, and I hope those newly added experiments, analysis, as well as practical guides can go to the camera ready version.

---

### Official Review · Reviewer_aWhg · 2025-07-03

**Clarity:** 3
**Significance:** 2
**Originality:** 2
**Rating:** 3
**Confidence:** 4

**Summary:**

This paper provides the first systematic study on which properties of preference data matter most for DPO in llm alignment. Through theoretical analysis and extensive experiments, the authors show that the absolute quality of chosen responses in the preference pairs plays a dominant role in DPO training, while the quality of rejected responses has only a limited impact once basic contrastiveness is ensured. The paper further reveals that increasing contrastiveness or mixing on-policy data is effective primarily because it raises the quality of chosen responses. These findings offer practical guidance for constructing high-impact preference datasets for DPO, highlighting the importance of curating high-quality positive responses to maximize alignment performance across diverse benchmarks.

**Questions:**

My questions are stated in the weakness section.

**Ethical Concerns:**

["NO or VERY MINOR ethics concerns only"]

**Final Justification:**

This paper provides a solid theoretical analysis and experiments. However, the main finding, while rigorously demonstrated, aligns with established intuition and could be perceived as lacking in significant novelty. Therefore, I leave my score as borderline accept.

**Limitations:**

Yes

**Paper Formatting Concerns:**

No concern

**Quality:**

2

**Strengths And Weaknesses:**

**Strengths**
- The paper provides the first systematic theoretical and empirical analysis of how preference data properties affect DPO, offering clear guidance on what truly matters for DPO-based LLM alignment.
- The findings deliver practical advice for data curation—demonstrating that improving the quality of chosen responses is the primary driver of performance, while the quality of rejected responses is far less important once minimal contrastiveness is achieved.
- The work demystifies common practices such as increasing the preference gap or on-policy mixing, showing that their main benefit comes from raising positive response quality.

**Weaknesses**
- The paper does not propose any new algorithm or model, focusing entirely on analysis rather than methodological innovation.
- The results are specific to DPO, and may not directly generalize to other alignment frameworks such as IPO, ORPO, SimPO, etcs.
- The paper highlights the importance of curating high-quality chosen responses but does not address the practical challenges of reliably collecting or evaluating such data in real-world settings.

---

> ### Author Rebuttal · Authors · 2025-07-29
>
> We sincerely appreciate your effort and your time in reviewing our paper. Please see the below for our responses.
>
> ### No new model/algorithm/methodological innovation
>
> We sincerely appreciate your encouragement to translate our observations into new methods or algorithms. However, we respectfully disagree with the premise that contributions must involve algorithmic novelty to be considered innovative. A core contribution of our paper lies in theoretical clarity and novel insights into the role of data in the behavior of DPO, a widely adopted yet not well understood method. Our work characterizes the impact of various preference data structures on DPO’s performance—results that were previously unknown and that address pressing open questions in the community. Foundational analysis has a long-standing tradition of driving progress by revealing when, why, and how existing methods work. We believe our contribution fits firmly within this tradition.
>
> ### Specific to DPO
>
> We fully acknowledge that our analysis focuses specifically on DPO. However, we believe this focus is well-justified for several reasons. First, DPO is one of the most foundational and widely adopted paradigms in preference-based fine-tuning. Many recent alignment methods are either inspired by or closely related to DPO. While our results are centered on DPO, they have broader implications for understanding and improving these related methods. Second, our work provides a clear and extensible framework—particularly in the numerical analysis—that can serve as a starting point for investigating the role of data in other alignment approaches. To further support this, we will release our codebase, which can be easily adapted to analyze or benchmark other alignment methods beyond DPO.
>
> ### Practical challenges in collecting high-quality data
>
> Thank you for raising this important and practically relevant point. Our paper is primarily analytical in nature, and its goal is to explain the impact of data quality on the performance of DPO. While implementation-level data curation strategies are important, they might be outside the scope of this work. However, we do identify concrete properties—such as the informativeness of chosen–rejected margins and the effect of suboptimal preference scores—that guide future data collection. Rather than offering ad-hoc heuristics, our framework provides principled criteria for evaluating whether a dataset is suitable for effective DPO training.

---

### Note · Authors · 2025-08-12

Dear AC and Reviewers,

Thank you for the thorough and insightful review process. The discussions have been incredibly valuable and have significantly strengthened our paper.

Our work provides a systematic study of **what matters in data for DPO**, demonstrating both theoretically and empirically that **the absolute quality of the chosen response is the primary driver of performance**. The quality of the rejected response and the preference gap, while relevant, play a secondary role.

The reviewer-author discussions were highly productive. Prompted by Reviewer jmf2's excellent suggestions, we conducted extensive new experiments, which are now part of our discussion. These include:

- A more granular analysis across five quality levels, confirming that improving chosen response quality yields consistent, monotonic gains, while varying rejected response quality does not.
- Additional controlled experiments that successfully disentangle the effects of preference gap and absolute quality, reinforcing that quality is the dominant factor.
- A direct numerical comparison between our online DPO setting and continual SFT, empirically supporting the claims in Theorem 4.5.
These new results have solidified our conclusions and addressed the reviewers' primary concerns.

The discussions also helped us clarify that our analysis of a specific online DPO variant (with fixed chosen responses) serves as a controlled experiment. It helps explain why general online DPO works well in practice: its success stems mainly from improving the quality of chosen responses, a non-trivial insight that challenges the community's traditional focus on contrastiveness.

We believe the revised paper offers the community clear, actionable guidance for constructing high-impact preference datasets in an efficient way. We are grateful for the opportunity to improve our work through this collaborative process with the reviewers.

Thank you all for your time, input and consideration.

---

### Decision · Program_Chairs · 2025-09-17

**Decision:**

Accept (poster)

**Comment:**

This paper investigates a fundamental question in alignment: which properties of preference data most influence DPO performance. The authors provide both theoretical analysis and comprehensive experiments. The central finding is that the absolute quality of the chosen responses dominates DPO outcomes, while the rejected responses and preference gap play only a secondary role once minimal contrastiveness is satisfied. The paper further shows that in an online variant with fixed chosen responses, DPO reduces effectively to SFT with mild regularization, helping interpret why online DPO often works.

Strengths:
- A timely and important contribution given the growing adoption of DPO.
- Careful theoretical development, including Theorem 4.5, that connects DPO to SFT.
- Extensive empirical work, including additional experiments suggested during review (granular chosen/rejected quality analysis, disentangling gap vs quality effects, and online DPO vs SFT comparisons). These strengthen the robustness of the conclusions.
- Clear practical takeaways for dataset construction in alignment pipelines.

Weaknesses:
- The main insight, that chosen response quality matters most, is consistent with intuition and some prior practice. The novelty lies more in its rigorous articulation and empirical confirmation rather than a surprising or methodologically new contribution.
- The scope is limited to DPO and a particular online DPO variant; generalization to other preference-learning methods (e.g., IPO, ORPO, SimPO) remains speculative.
- Some reviewers felt the contribution was narrower than initially expected and that quantitative measures of relative importance could be stronger.

The authors responded constructively to reviewer concerns, adding substantial new experiments that directly addressed calls for more granular analysis and for empirical support of the theoretical claims. Two reviewers increased their scores to accept after these revisions. Another reviewer remained cautious, viewing the conclusions as intuitive and narrow in scope, but did not strongly contest the technical soundness.

Overall, the paper makes a clear, well-executed, and practically useful contribution to understanding DPO, even if the conceptual novelty is moderate. The theoretical clarity and expanded experimental support justify merits. The work will be of interest to practitioners building alignment datasets and to researchers studying preference-based learning.